# Decoding health disparities by gender, ethnicity and chronic diseases across three Latin American countries

Carlos Chivardi [1] ✉, Alejandro Zamudio Sosa [2],
Daniella Medeiros Cavalcanti [3], José Alejandro Ordoñez [4], Cristina Almeida[5],
Philipp Hessel [6], Ana L. Moncayo [7] & Davide Rasella [3,8]

Chronic diseases disproportionately affect certain ethnic and gender groups, but the social determinants driving these disparities in Latin America are not fully understood. In this study, we analyzed data from national health surveys in Brazil, Mexico, and Ecuador (2018–2019), representing a total weighted population of 96,726,891 adults. We used random forest models to predict chronic disease diagnoses based on education, occupation, and access to essential services such as sanitation, drinking water, and garbage collection. Our models performed better for indigenous and afro-descendant people, highlighting significant inequalities. While occupation and education were strong predictors for women, decreasing model performance by 8.57% and 7.36% respectively, occupation was the critical variable for men, decreasing model performance by 19.6% when neutralized. This work highlights the need for public policies adapted to the specific needs of different ethnic and gender groups.

Gender has emerged as a central element in understanding health disparities, defined as "the set of socially constructed roles and relationships, personality traits, attitudes, behaviours, values, relative power and influence that society differentially attributes to the two genders"[1]. Gender influences health policies, biomedical and contraceptive technologies, and health system responses[2]. These disparities extend beyond biological differences, encompassing social, cultural, behavioural, and biological factors. Social scientists note that biological differences alone cannot explain health outcomes; social determinants and gender play a crucial role[2].

Gender-based health disparities often relate to behaviours associated with gender roles and it has been found that such roles can strongly influence different health outcomes in Latin America[3,4]. Men typically exhibit risk behaviours, such as increased cigarette and alcohol consumption, while women more frequently use health services and monitor physiological changes[5]. For instance, women are more likely to seek primary medical care for illness symptoms[6]. However, gender differences in chronic disease morbidity are sensitive to sample characteristics and analysis methods and vary by country due to cultural and social factors[5,7]. Differences in health outcomes are linked to power, autonomy, poverty, and marginalization[8].

On the other hand, there is a wide literature that indicates that in addition to gender, ethnicity has also been related to various health outcomes (infant mortality, child nutrition, maternal health, reproductive health, rheumatic diseases, etc.) that differentially affect ethnic groups in Latin America, with people of indigenous descent being the most vulnerable than any other ethnic group[9].

[1]MSc, Centre for Health Economics, University of York, York, UK. [2]PhD, School of Psychology, National Autonomous University of Mexico (UNAM), Mexico City, Mexico. [3]PhD Institute of Collective Health, Federal University of Bahia (UFBA), Salvador, Brazil. [4]Department of Statistics, Pontifical Catholic University of Chile, Santiago, Chile. [5]MSc Center for Health Research in Latin America (CISeAL), Pontifical Catholic University of Ecuador, Ecuador, Ecuador. [6]PhD University of Los Andes, Bogotá, Colombia / Swiss Tropical & Public Health Institute, Basel, Switzerland. [7]Centro de Investigación para la Salud en América Latina (CISeAL), Pontificia Universidad Católica del Ecuador, Quito, Ecuador. [8]PhD Institute of Global Health (ISGlobal), Hospital Clinic, Universitat de Barcelona, Barcelona, Spain. ✉e-mail: carlos.chivardi@york.ac.uk

**Table 1 | Descriptive analysis by gender and ethnicity group**

| | Men | | | | Women | | | | |
| --- | --- | --- | --- | --- | --- | --- | --- | --- | --- |
| | Mixed | Black | Indigenous | Others | Mixed | Black | Indigenous | Others | *p*-value |
| **Dependent variable** | | | | | | | | | |
| Dx chronic disease | 0.12 | 0.15 | 0.09 | 0.18 | 0.15 | 0.21 | 0.11 | 0.19 | > 0.001 |
| **Independent variables** | | | | | | | | | |
| Piped water | 0.88 | 0.97 | 0.57 | 0.98 | 0.87 | 0.97 | 0.53 | 0.98 | > 0.001 |
| Water | 0.94 | 0.98 | 0.77 | 0.98 | 0.94 | 0.98 | 0.78 | 0.98 | > 0.001 |
| Sewage | 0.93 | 0.95 | 0.76 | 0.96 | 0.93 | 0.96 | 0.77 | 0.96 | > 0.001 |
| Garbage | 0.9 | 0.92 | 0.66 | 0.94 | 0.91 | 0.95 | 0.64 | 0.96 | > 0.001 |
| Urban location | 0.81 | 0.88 | 0.51 | 0.89 | 0.82 | 0.9 | 0.51 | 0.92 | > 0.001 |
| **Occupation** | | | | | | | | | |
| Informal employment | 0.19 | 0 | 0.46 | 0 | 0.13 | 0.00 | 0.25 | 0 | > 0.001 |
| Formal employment | 0.49 | 0.59 | 0.33 | 0.53 | 0.30 | 0.40 | 0.18 | 0.39 | |
| Retiree | 0.15 | 0.29 | 0.03 | 0.39 | 0.19 | 0.44 | 0.05 | 0.5 | |
| Unemployed | 0.14 | 0.10 | 0.16 | 0.06 | 0.36 | 0.14 | 0.50 | 0.1 | |
| **Education** | | | | | | | | | |
| No Education | 0.31 | 0.38 | 0.48 | 0.29 | 0.33 | 0.40 | 0.52 | 0.32 | > 0.001 |
| Education primary | 0.24 | 0.17 | 0.27 | 0.13 | 0.23 | 0.13 | 0.29 | 0.1 | |
| Education secondary | 0.28 | 0.36 | 0.18 | 0.36 | 0.26 | 0.34 | 0.12 | 0.32 | |
| Education Higher | 0.16 | 0.08 | 0.06 | 0.20 | 0.16 | 0.11 | 0.05 | 0.24 | |
| Age | 45.31 (17.37) | 46.62 (17.6) | 45.77 (17.1) | 50.42 (18.4) | 46.93 (17.7) | 50.35 (18.66) | 46.19 (17.7) | 53.09 (18.8) | > 0.001 |

All variables represent proportion, except for age where the mean and standard deviation are shown. One-sided chi-square tests are used to determine if there are statistical differences between genders, and two-sided Kolmogorov-Smirnov tests are used for age distribution. Source data are provided as a Source Data file.

In examining the gender-ethnic intersection, social determinants show differentiated roles. Szanton et al.[10] found income and education increased morbidity and mortality risk in older women, regardless of race. Assari et al.[11] reported that education lowered health problems for whites but not blacks, and income protected women more than men. Reducing early life adversities decreased cardiovascular disparities more among men of different ethnicities than among women[12]. Social inequalities like socioeconomic status have differential effects on health at the gender-ethnic intersection[13]. However, although gender and ethnicity may play important roles as social determinants, they are not the only ones. Social conditions such as lifestyles, living and working situations, neighbourhood characteristics, poverty, environmental pollution, income, education and occupation have been widely studied as social determinants of different chronic diseases. In this sense, income, education and occupation characteristics have been shown to be the most important social determinants[14], followed closely by inequalities associated with ethnicity[15]. Despite this, the meaning and direction of social determinants such as education, occupation and living conditions are consistent even without taking into account population and gender; people with a lower level of education, informal jobs or no jobs, and those lacking basic services are significantly more likely to suffer from chronic diseases and have worse treatment expectations[14]. According to Cockerham[14], the above-mentioned variables have indirect effects on chronic diseases by influencing lifestyles, health behaviors, food security, stress levels as well as biological outcomes through genetic expression and generational inheritance. Health researchers emphasize the intersection of race/ethnicity, class, income, education, age, and geography in addressing health inequalities[16]. In developed countries, black individuals and women have higher chronic disease rates than their white counterparts[17,18]. In Latin America, darker-skinned individuals, especially black and indigenous women, have worse health outcomes[19,20]. Despite extensive research on gender and ethnicity, there is limited intersectional analysis in Latin America, particularly regarding multiple chronic diseases. Furthermore, although the relationship between social determinants and the most prevalent chronic diseases in Latin American countries has been widely studied, there is little evidence of possible non-linear relationships and differences in these relationships between ethnic and gender groups. Few studies use machine learning models to explore social determinants of chronic diseases by gender and ethnicity in Latin America. Machine learning can reveal linear and non-linear relationships and often outperforms traditional models[21]. Our study aims to evaluate the predictive role of social determinants by gender and ethnicity in the diagnosis of chronic diseases (diabetes, cardiovascular diseases, kidney diseases, cerebrovascular diseases and obesity), which are among the leading causes of mortality in three (Brazil, Ecuador, and Mexico) countries of Latin America[22].

## Results
### Descriptive results
Table 1 presents the results of the descriptive analysis delineating proportions and means with standard deviations across gender and ethnicity groups. The chi-square test was used for categorical variables, and Kolmogorov–Smirnov for age distribution. Concerning diagnosis of chronic diseases, a higher proportion was observed in black women (0.21) and other ethnicities in men (0.18). Among men, those identifying as others exhibited higher proportions in variables such as access to piped water (0.98), water supply (0.98), and sewage disposal (0.96), similar to men who identified as black. Similarly, women of others ethnicity demonstrated relatively higher proportions in these categories. The variable urban location also showed significant distinctions, with higher proportions in the black and others ethnicity categories. Occupation-wise, black individuals tended towards formal employment, while those of indigenous demonstrated a higher proportion in informal employment. Educational disparities were evident, with indigenous and black having lower proportions in higher education. The mean age varied significantly across gender and ethnicity groups, with black men having a mean age of 46.62 (SD = 17.6) and

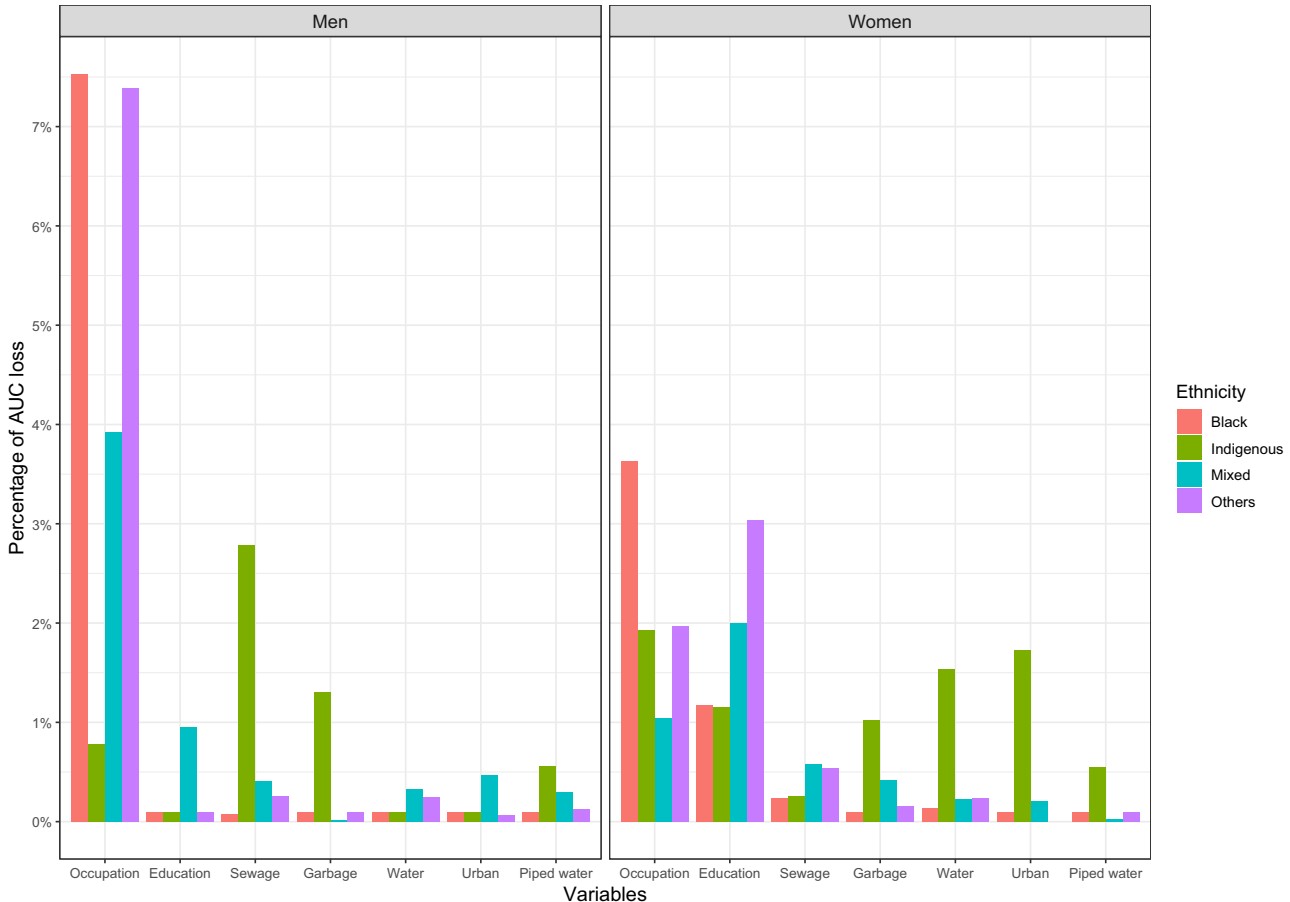

**Fig. 1 | Importance of variables in predicting the diagnosis of chronic diseases according to gender and ethnicity for the random forest model.** The graph shows the importance of variables in predicting chronic diseases in each ethnic group and by gender. The importance of the variables is obtained using an agnostic approach that iteratively neutralizes each variable through sampling and measures the area under the curve lost when neutralizing each variable individually. The models were trained using the variables age and country; however, these variables are not shown in the graph because they were added as control variables. Source data are provided as a Source Data file.

black women at 50.35 (SD = 18.66). Statistical significance was observed across all variables ($p < 0.001$). Descriptive analyses by country are provided in TableS3–S5 of the supplementary material.

## Machine learning results

Figure 1 illustrates the importance of various social determinants in predicting chronic disease diagnoses, differentiated by gender and ethnicity. The importance is measured by the percentage of area under the curve (AUC) lost if each variable is neutralized from the model. Overall, education and occupation were found to be more significant determinants than access to public services. For both genders, occupation was the most important variable for predicting chronic diseases; however, for men, occupation was more critical, with an AUC loss of 19.62% compared to 8.57% for women. In contrast, education was much more important for women, resulting in a total AUC loss of 7.36% compared to 1.25% for men.

For the indigenous population, the occupation was the most important variable for women (1.93% AUC loss) followed by urban location and men was the sewage (2.78% AUC loss), followed by garbage and occupation. In the black population, occupation was the most important variable for women and men (3.63% and 7.53% AUC loss respectively) and education was the second most important, mainly for women (1.17% AUC loss). Among mixed-ethnicity individuals, education was determinant most significant factor for woman (2% AUC loss) and the second for men (0.95% AUC loss), occupation also was determinant for both women (1.04% AUC loss) and men

(3.92% AUC loss) being more important for men. For individuals of other ethnicities, education was most important for women (3.04% AUC loss) and occupation form men (7.93% AUC loss). The most considerable ethnic differences were observed in the importance of occupation for black women (3.63% AUC loss) compared to mixed ethnicities women (1.04% AUC loss) and in occupation for black men (7.53% AUC loss) compared to indigenous men (0.78% AUC loss). Regarding the country's predictive significance, it was most significant among indigenous women (7.74% AUC loss), followed by mixed men (6.59% AUC loss). Meanwhile, it was least significant among men of the "other" ethnic group and black men (2.47% and 1.47% AUC loss, respectively). Further details are provided in Table S2.

Figure 2 illustrates gender differences in AUC loss by ethnicity. Across all ethnic groups, occupation consistently shows the largest disparity between gender. In the others and black population, the gender gap is most pronounced in occupation, with a difference of almost 5.42 and 3.9 percentage points respectively. Among indigenous individuals, the sewage again stands out, with a 2.52-point difference. For the mixed-race population, gender differences are also highest in occupation, with a nearly 2.88-point gap. While occupation consistently emerged as the key factor across ethnicities, education also shows differences, particularly in "other" groups. Additionally, infrastructure-related variables such as piped water, urban living, garbage collection, and sewage access show smaller, yet still important, disparities, especially in the indigenous population. Regarding the sensitivity analyses by country, the results agree with the general

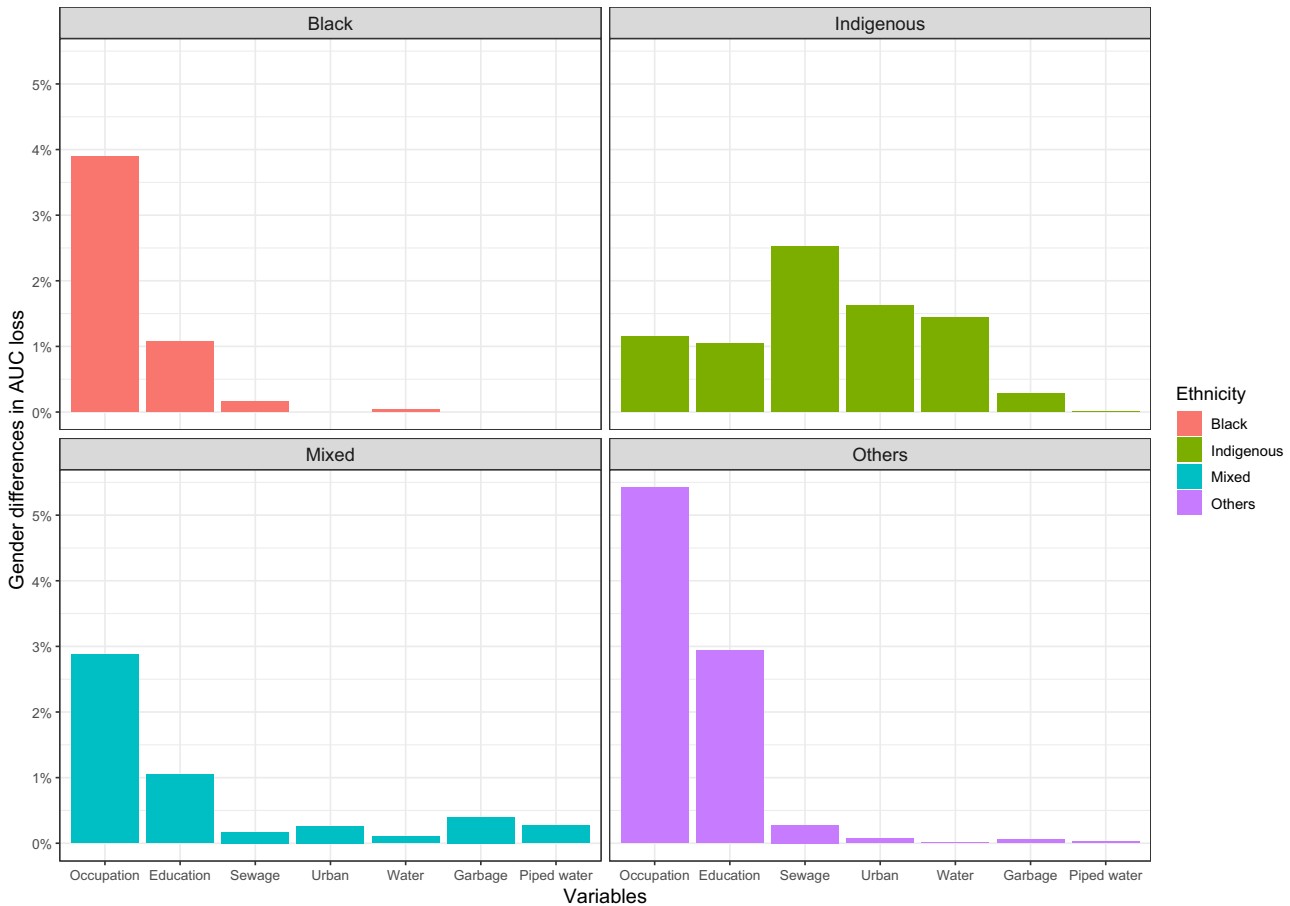

**Fig. 2 | Difference in AUC lost between genders for each ethnicity group.** This graph shows the difference in the importance of variables between genders in each ethnic group in each of the models trained by subgroup, so that the greater the percentage of the variable in the area under the curve that is missing, the greater the difference in the importance of that variable between men and women. The models were trained with the variable age and country; however, these variables are not presented in the graph since they were added as control variables. Source data are provided as a Source Data file.

results. Specifically, it was found that in all countries, the majority of variables considered in the present study were more important for the black population and the indigenous population.

Figure 3 presents the cumulative partial dependency plots for men, highlighting the relationship between various social determinants and the probability of being diagnosed with a chronic disease across different ethnic groups. Notably, a higher level of education is associated with a higher probability of diagnosis of chronic diseases, particularly in the indigenous population and other populations, where the probability increases from 0.03 without education to 0.05 with higher education in indigenous populations and from 0.16 and 0.18 in other populations respectively. This trend is less pronounced for black group but follows a similar pattern. Occupation exhibits an inverse relationship, with the highest probability of diagnosis among retirees and the lowest among those with casual and formal employment across all groups. The services like access to piped water, sewage and water generally reduce the probability of diagnosis, but the other and indigenous populations show a distinct pattern: their probability of chronic disease remains steady or even increases with greater access to these services. Age also plays a role, with diagnosis probability rising sharply as age increases.

Figure 4 presents the cumulative partial dependency plots for women, the influence of education and occupation on the likelihood of chronic disease diagnosis across various ethnic groups was really important. Unlike men, for women, higher education levels are generally associated with a decrease in the probability of being diagnosed

with a chronic disease. This trend is particularly evident in black and "other" ethnic groups, where the probability decreases from 0.14 and 0.11 for those with no education to 0.23 and 0.12 for those with higher education, respectively. In terms of occupation, retiree is linked with higher probabilities of diagnosis across almost all ethnic groups. Conversely, formal employment generally correlates with lower probabilities, mirroring patterns seen in men. Regarding drinking water, garbage collection, and sewage treatment, the effects vary depending on the entity. However, as in the case of men, indigenous women experience a more complex pattern, where the probability remains constant or even increases despite improved access to these services, underscoring potential disparities in how these factors impact different populations. The age variable shows an increasing trend, with chronic disease likelihood rising steadily as women age.

## Discussion

While the relationships between social determinants and chronic diseases have been widely studied in Latin America, significant gaps remain in the literature. First, most previous studies have relied on traditional regression-based approaches, which may not fully capture the complex, nonlinear interactions between social determinants. Second, social determinants are well-studied, few studies in the region have explored their differential impact across genders. Third, there is limited research on how social determinants interact specifically within indigenous and black populations, despite these groups facing some of the greatest health inequities. By addressing these gaps, our

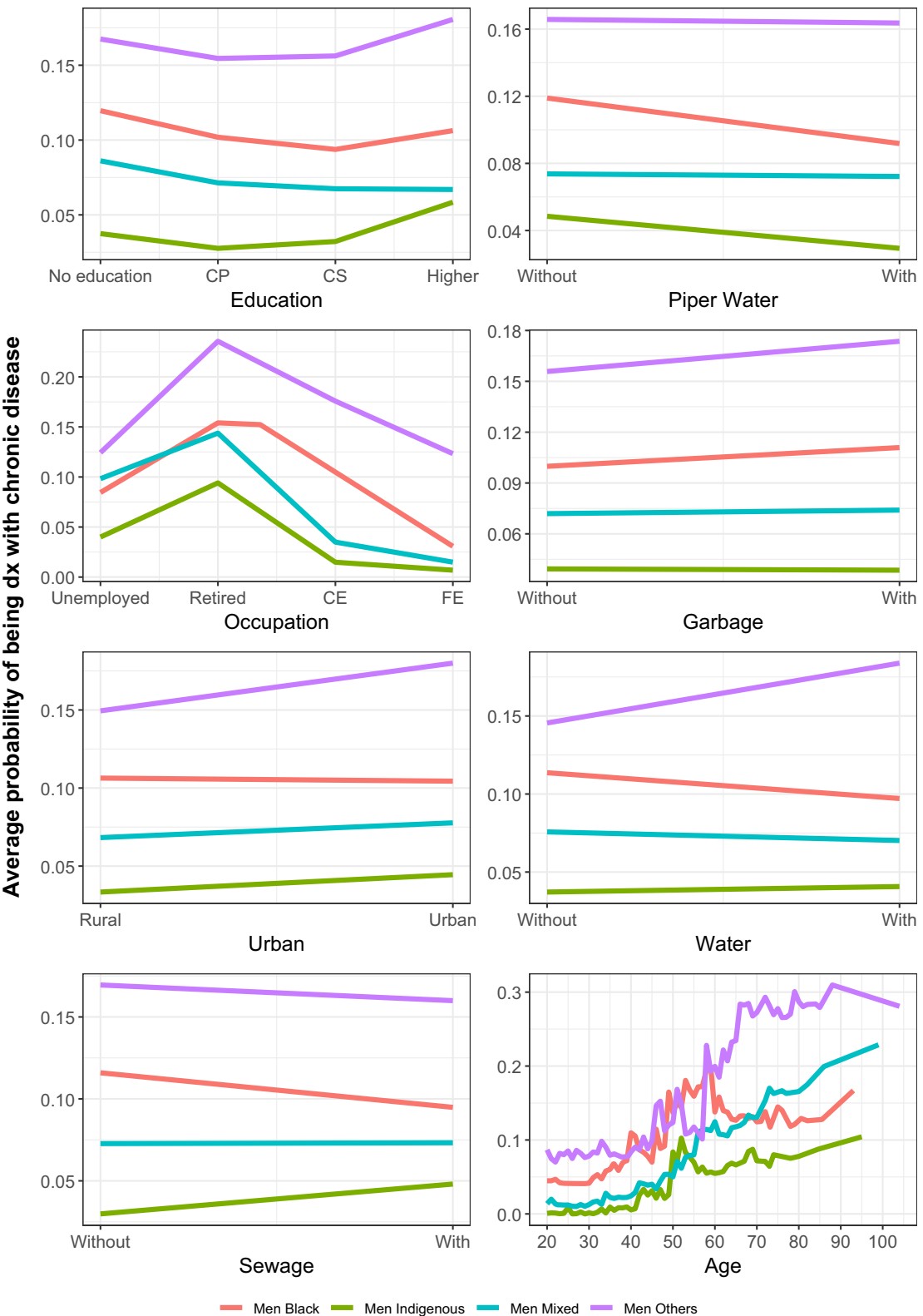

**Fig. 3 | Cumulative local profiles for men.** The graph shows, for each variable considered, the average probability change that the random forest model calculated for having a chronic disease as a function of the change in each variable for men. The difference in the probability ratio by ethnic group is shown for each variable. For the education variable, CP means completed primary school, and CS means completed secondary school. For the occupation variable, CE means casual or informal employment, and FE means formal employment. Source data are provided as a Source Data file.

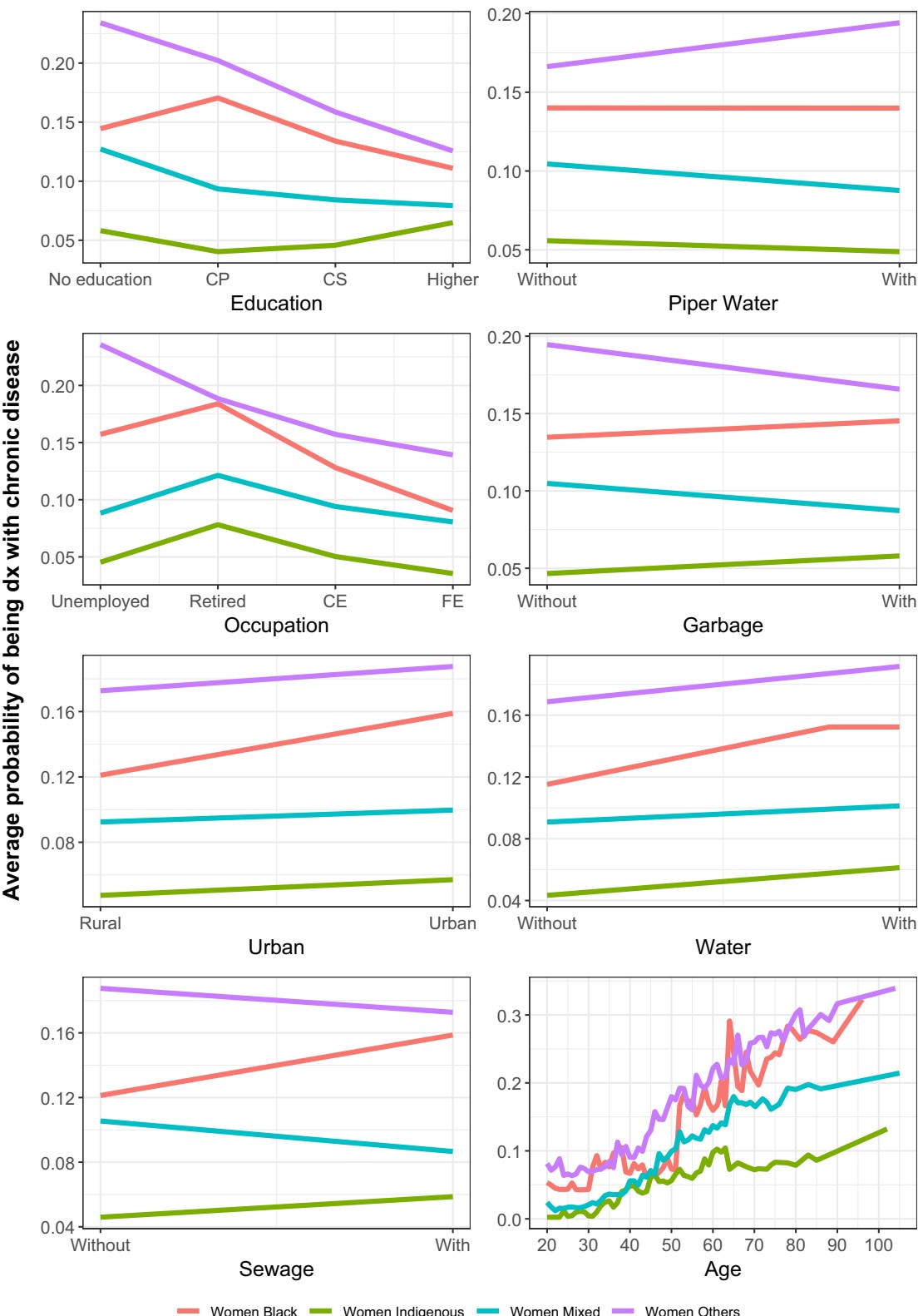

**Fig. 4 | Cumulative local profiles for women.** The graph shows, for each variable considered, the average probability change that the random forest model found as a function of the change in each variable for women. The difference in the probability ratio by ethnic group is shown for each variable. For the education variable, CP means completed primary school, and CS means completed secondary school. For the occupation variable, CE means casual or informal employment, and FE means formal employment. Source data are provided as a Source Data file.

study provides evidence on the role of social determinants in health disparities in Latin America and supports specific policy interventions to reduce these inequities.

We found that the indigenous population faced the worst socio-economic conditions, with men particularly disadvantaged compared to women. Conversely, individuals of mixed ethnicity had the best socioeconomic conditions across most social determinants. Regarding the results of the random forest models, the indigenous population was where the models showed the best performance (AUC = 0.83 for women, AUC = 0.86 for men), followed by the mixed ethnicity population (AUC = 0.81 for women, AUC = 0.82 for men), then the black population (AUC = 0.78 for women, AUC = 0.76 for men) and finally the other ethnicity population (AUC = 0.75 for women, AUC = 0.78 for men). In other words, the trained models showed the highest AUC for predicting chronic diseases among the indigenous. This suggests that the social determinants considered were particularly impactful for indigenous population, and the machine learning models effectively captured the complex interactions of these determinants. Factors such as lack of education and non-formal employment are likely linked to reduced access to healthcare, increased stress, lower food quality, and poorer self-care, which contribute to chronic disease risk, especially in women. Additionally, inadequate home conditions like lack of piped water, sewage, or garbage collection increase exposure to risk factors, further predicting chronic diseases. Choo and Ferree[23] suggest that multiple structural disadvantages create an interlocking system of oppression that limits opportunities for marginalized groups. Thus, the combined effects of ethnicity and gender-related disadvantages likely enhance the predictive power of the social determinants considered. The use of cross-validation and test set evaluation minimized the risk of overfitting in these models.

In terms of gender, women exhibited higher proportions of diagnosis disease than men, aligning with existing literature findings[24,25]. This observed trend could potentially be attributed to greater utilization of health services or seeking help, a pattern commonly observed with women using services more frequently than men. Another plausible explanation is the variance in sensitivity and monitoring of physiological changes, with women generally being more attentive to such changes, as indicated by Gordon and Hubbard[5].

When we analysed the differences between genders within the same ethnic groups, we found some patterns. Firstly, education consistently played a more significant role for women across all ethnic groups. Notably, among "others" and mixed populations, substantial disparities favored women as a crucial social determinant. An explanation for this may be that education in women potentiates the positive effect of self-care patterns, constant monitoring of physiological changes, and attention-seeking those women already present at higher rates compared to men[5]. Third, the significance of occupation as a social determinant in predicting chronic diseases was notably higher for men than women across the rest of ethnic groups. A possible reason could be that in the case of men, having a formal job reduces the presence of psychological stress, possibly associated with the role of provider, which would reduce the probability of developing a chronic disease[26,27]. However, other studies have found that employees with chronic illnesses are more likely to leave paid employment compared to healthy employees[28]. More studies are necessary to evaluate the possible mediating role of psychological stress associated with the work condition and its relationship with the development of chronic diseases. Another possible explanation is that men who have formal employment are more likely to have food security and better-quality diet compared to their counterparts with informal jobs. Studies confirm the strong association between food security and the presence of chronic diseases such as hypertension, coronary heart disease, diabetes, kidney disease, etc[29].

When examining social disparities among women, the most significant difference was observed in the area of occupation, being more influential for black women and less significant for women of mixed ethnicity. Similarly, the most significant divergence among ethnicities for men was observed in the domain of occupation, where it held the utmost importance for black men and the least for indigenous men. Notably, indigenous men exhibited inferior social conditions compared to their black counterparts, characterized by a lower proportion of essential facilities like piped water and sewage.

Additionally, we observed in the accumulated-local profiles that in the indigenous population the probabilities of being diagnosed with a disease were inverse in the majority of determinants used, and at the same time it was the models of these populations that had the best performance in terms of the AUC, this may be because the accumulated-local profiles "control" the correlations between predictors and show the changes in unique probabilities of the variables without counting the interactions. It is possible that the interactions between the social determinants are more important in indigenous populations, perhaps the models within indigenous populations detected latent patterns in the interaction of the determinants, it is necessary to continue exploring these relationships mainly in this population. A similar pattern was found in the "other" population in several of the determinants, however in the importance of the variables these were found to be of little relevance for the prediction, so this prediction relationship has little or no effect on the prediction of the chronic diseases considered.

A limitation of our research was the use of self-reported diagnosis; there is evidence of reasonable agreement between self-report and medical records[30], but the accuracy of self-report may be affected by factors such as severity and acuity of the disease and the possible overestimation of women or underestimation of men[5]. Future studies should consider using multiple sources of information to determine the prevalence (rather than purely self-report methods) to reduce potential reporting bias. Another important limitation is that ethnic classification is imperfectly harmonised across surveys. In Mexico, ENSANUT 2018 does not capture self-identified ethnicity; we use "speaks an Indigenous language" as a proxy for Indigenous identification. This proxy may misclassify individuals and does not identify afro-descendant people. As a result, cross-country contrasts involving Afro-descendant groups are limited to Brazil and Ecuador, and Mexican results should be interpreted as "Indigenous-language speakers vs other" rather than a comprehensive ethnicity taxonomy.

A key limitation is imperfect outcome harmonisation across surveys. Mexico and Brazil use "ever diagnosed" items, whereas Ecuador provides a 30-day report; these differences in recall window and wording mean cross-country prevalence levels are not directly comparable. We mitigate this by focusing interpretation on within-country patterns (gender/ethnicity strata) and by including country indicators in pooled models, but residual non-comparability is possible. Another limitation is that we do not adopt biomarker-defined outcomes in the main pooled analysis because comparable biomarker data are not available with full adult coverage in the same wave for all three countries, and relying on subsamples would substantially reduce effective sample sizes and introduce selection differences across settings. Future extensions will incorporate additional surveys and waves (including those with harmonised biomarkers) when they jointly provide nationally representative coverage of both objective health measures and the social-determinant variables required for our comparative framework.

A further limitation is that the surveys capture sex (male/female) rather than gender identity or gendered social roles; accordingly, we use recorded sex as a proxy for gender. This proxy may introduce misclassification and excludes transgender, non-binary, and other gender-diverse individuals, potentially biasing subgroup contrasts; the direction and magnitude of any such bias are uncertain and may vary by country and ethnicity.

Relatedly, all outcomes are self-reported and thus susceptible to differential access to diagnosis, awareness, and reporting bias across subgroups (e.g., by education or service access). The net direction of any resulting bias is uncertain. Future work linking to clinical records or using harmonised biomarker-based definitions would improve comparability and strengthen causal interpretation, and incorporating harmonised measures of sex assigned at birth, current gender identity, and lived gender roles (with inclusive categories) would improve construct validity and inclusion.

Another limitation of our study is the failure to include other determinants in the social environment such as discrimination or even the language spoken. Castro, Savage and Kaufman[31] consider that an important factor for women to have worse health outcomes is due to discrimination on the part of health providers. This discriminatory factor has already been reported in other studies in LA[32]. The real and perceived low economic level associated with dark-skinned people can also be a reason for discrimination by health service providers[19]. Furthermore, our focus on three countries reflects the predefined scope and feasibility of the data; therefore, external generalizations beyond these settings should be made with caution until comparable data from other countries are incorporated. Despite the limitations found, our study has several strengths. First, the use of multiple populations from different countries helped us take into account possible cultural differences in terms of gender and ethnicity and measure their role as a social determinant. Secondly, the use of machine learning models, with cross-validation, allowed us to evaluate and select the models with the best performance to extract information from them. Third, our sensitivity analysis confirmed several of our main results, such as that the variables considered were more important in predicting vulnerable populations and those with structural injustices such as the indigenous and black populations.

Our study highlights the importance of analysing social determinants as an interconnected system rather than in isolation, as demonstrated by the strong predictive performance of our machine learning models. A key finding is the differential role of education and employment across genders, with education being more critical for women—likely due to its impact on health literacy and preventive care—while employment status plays a stronger role for men, potentially due to economic stability and healthcare access. This gendered effect, which has been poorly documented in the literature, underscores the need for targeted health policies that prioritize education for women and employment-related interventions for men. Additionally, our results reveal that social determinants have a stronger and more complex influence among indigenous population, with interactions between multiple factors playing a more significant role compared to mixed and other ethnic groups. Notably, higher education was paradoxically associated with an increased likelihood of chronic disease diagnosis among indigenous individuals, possibly reflecting improved healthcare access leading to higher detection rates rather than increased disease burden. Another possible explanation is that higher education provided this population with greater purchasing power and this in turn reflects changes in health behaviours in nutrition and substance use[33]. Furthermore, infrastructure access (piped water, sewage, and garbage collection) emerged as a significant yet unevenly distributed determinant, disproportionately affecting indigenous women. Importantly, much of the research on social determinants of health (SDH) focuses not only on identifying whether specific determinants exert an effect, but also on quantifying the magnitude of these effects. Traditional statistical models often require strict assumptions about linearity and variable independence, which may obscure the true magnitude of influence, particularly when determinants interact in complex, non-linear ways. In contrast, machine learning models excel in capturing these intricate relationships without imposing rigid model structures. This flexibility allows for more accurate and nuanced estimation of effect sizes across diverse population subgroups, as

demonstrated by the strong and interpretable patterns observed in our analyses. By leveraging machine learning, our study moves beyond binary associations to provide a clearer picture of how multiple, interrelated SDH factors jointly shape health outcomes. These findings emphasize the need for comprehensive public health strategies that address both social and environmental determinants. Policymakers should implement education-based interventions for women, employment-focused policies for men, and holistic infrastructure improvements for indigenous and black communities to mitigate structural inequalities and improve health outcomes. These findings emphasize the need for comprehensive public health strategies that address both social and environmental determinants. Policymakers should implement education-based interventions for women, employment-focused policies for men, and holistic infrastructure improvements for indigenous and black communities to mitigate structural inequalities and improve health outcome.

The implications of our study for public policy are substantial, emphasizing the need for nuanced interventions to address health disparities associated with gender and ethnicity. Policymakers must adopt targeted strategies that acknowledge the intricate interplay of these factors. The distinct socioeconomic challenges faced by the indigenous population, particularly men, highlight the imperative for tailored policies that respond to their unique circumstances. The enhanced predictive performance of machine learning models in chronic disease outcomes among indigenous and black populations underscores the pivotal role of social determinants in shaping health disparities within these communities. To effectively tackle these disparities, public policy initiatives should prioritize educational interventions for women and employment-related strategies for men. Empowering women across ethnicities through increased access to education can amplify positive self-care patterns and health behaviours. Simultaneously, interventions aimed at improving employment conditions for men, especially in non-indigenous populations, can contribute to stress reduction, increased healthcare access, and improved nutritional quality – essential factors in preventing chronic diseases.

Recognizing the pronounced impact of age on health outcomes, particularly among indigenous women and men, suggests the need for policies that consider the cumulative effects of structural disadvantages over time. Targeted efforts to improve working conditions for black men can swiftly translate into better access to essential resources, addressing fundamental conditions contributing to health disparities.

## Methods
### Study design
Our research complies with the relevant ethical standards and since it uses exclusively data from public and open surveys, explicit approval from an ethics committee beyond that of the funding agency was not required. The study objective was to analyse Mexico, Brazil and Ecuador; surveys were included if they provided (i) nationally representative adult microdata from 2018 to 2019, (ii) self-identified ethnicity/race with indigenous and afro-descendant categories, (iii) harmonisable indicators of chronic disease and key social determinants, and (iv) public or timely licensable access.

We used a multi-country individual-level cross-sectional study with data from Mexico, Brazil and Ecuador. We extracted individual-level data from the National Health Survey of Mexico ("Encuesta Nacional de Salud y Nutrición" – ENSANUT, 2018), the National Health Survey of Ecuador (ENSANUT, 2018) and the National Health Survey of Brazil ("Pesquisa Nacional de Saúde" – PNS, 2019. We use data from all adults over the age of 20 to standardize all surveys. The dependent variable is a binary indicator of chronic disease presence at interview, harmonised across surveys as follows: for Mexico and Brazil, we used self-reported clinician diagnosis (ever) of diabetes, cardiovascular

disease, chronic kidney disease, cerebrovascular disease or obesity in all three countries. For Ecuador, the closest analogue available in the 2018 wave is the "last-30-days" report of selected chronic conditions; we coded a positive report as having the condition at interview. Although ENSANUT Mexico includes laboratory modules (e.g., glycaemia/HbA1c) and PNS measures blood pressure, comparable biomarker data are not uniformly available for the same wave and full adult sample across all three countries, and laboratory modules are typically fielded on subsamples. To preserve cross-country comparability and statistical power in gender/ethnicity strata, our primary analyses therefore use the harmonised self-report definition. All pooled models include country indicators to absorb level differences attributable to measurement. The actual number of respondents included in our analysis was 42,068 for Mexico, 96,111 for Ecuador, and 40,088 for Brazil.

## Analytical variables

Our dependent variables were the self-reported diagnosis of chronic diseases (by a health professional for the case of Mexico and Brazil) of any of the following categories: (a) diabetes, (b) cardiovascular disease, (c) kidney disease, (d) cerebrovascular disease, and (d) obesity. In the case of Ecuador, chronic diseases such as high blood pressure, diabetes, obesity, cancer, arthritis and cardiovascular disease reported in the last 30 days at the time of the survey were considered, given the structure of the ENSANUT 2018 survey in Ecuador.

In this way we obtained a dichotomous variable for each adult that represented whether or not each adult was diagnosed with any of the chronic diseases listed. For the independent variables, we obtained data at the individual level on whether the person's home had water, sewage, garbage collection, what occupation (or employment status) they had (formally employee, informal or casual employee, unemployed, and retired), and education (without education, basic level, intermediate level or secondary and higher level). Additionally, given the important predictive role of age (as a continuous variable) and country (as a categorical variable) in the diagnosis of chronic diseases, we included age and country as control variables in the models. Since we focused on social determinants, these control variables were not included in the variable importance graphs but their importance is reported.

We analysed the social determinants stratifying groups by gender and ethnicity, in this way, we formed eight groups: mixed-ethnicity men, Indigenous men, black men, men of other ethnicities (mulatto, montuvio, whites and "others" from Ecuador), mixed-ethnicity women, indigenous women, black women and other ethnic women (mulatto, montuvio, whites and "others" from Ecuador). In the three surveys we used, gender was obtained from the self-report of the sex of each respondent.

To ensure consistency across the three national health surveys (ENSANUT 2018 in Mexico, PNS 2019 in Brazil, and ENSANUT 2018 in Ecuador), we standardized key variables related to chronic disease diagnosis and social determinants. Meanwhile, socioeconomic determinants (education, occupation, and access to essential services) were measured using equivalent categories across all data sets. Any discrepancies in classification were resolved by aligning responses with the most commonly used categories. The questions and criteria used to measure diagnosed diseases and independent variables can be found in Table S1 in the supplementary information.

## Statistical and machine learning analysis

We conducted a descriptive analysis to estimate the proportion or mean and standard deviation of each social determinant for eight population groups, using the expansion factors for each survey. To assess the role of social determinants in predicting the risk of chronic diseases, we performed the following steps for each subgroup: a) we trained a random forest[34] machine learning model by cross-validation

(10 repetitions, k-fold = 3). We use the expansion factors directly in the training of the algorithms and in cross-validation, and also directly in testing and model-agnostic approximation. We used the caret library[35] for cross-validation and the ranger library[36] to train the models with the best hyperparameters found. All analyses were conducted using R free software version 4.4.0. Random forest, proposed by Breiman (2001), is an ensemble of decision trees based on the principle of bagging and random attribute selection. It is formally defined as follows:

## Model construction

Let $D = \left\{ (X_i, Y_i) \right\}_{i=1}^{N}$ be a data set with $N$ observations, where $X_i \in R^p$ represents a feature vector and $Y_i$ the response variable.

1. Bootstrap sampling: $T$ subsets $D^{(t)}$ are generated by sampling with replacement of $D$.
2. Tree growing: For each subset $D^{(t)}$, a decision tree $h(X, \Theta_t)$, is built, where $\Theta_t$ represents the random selection of features at each node. Unlike classical trees, each split is performed with a randomly chosen subset of $m < p$ features.
3. Aggregate prediction (classification problem) majority voting is used:

$$\hat{Y} = \text{mode}\{h(X, \Theta_t) : t = 1, \ldots, T\}$$

We used 70% of the data for training and varying hyperparameters (*mtry* from 3 to 10 for the random forest) to obtain the model with the best performance and 30% to evaluate the performance of each model, b) we obtained the area under the curve (AUC) to evaluate the performance of the model to predict chronic diseases, c) we obtained the importance of the variables using the agnostic approach[37] and calculated the missing AUC by eliminating each variable from the models and, d) we obtained the accumulated local-dependence graphs to know the average probability of prediction of having a dx chronic disease based on the different values of the predictors taking into account the correlation between predictors. We use the random forest algorithm since it has proven to be robust and has good performance in the field of health research[38]. We selected the AUC as a performance metric because it has several advantages over other metrics in supervised classification algorithms: a) it is robust to imbalanced datasets where there are many cases of one category and few of another, b) it provides a global assessment of the model performance across the range of possible threshold values, facilitating comparison between different models, c) it has an intuitive interpretation where 1 indicates perfect performance, while a value of 0.5 suggests performance equal to random classification[37]. On the other hand, the agnostic approach does not assume specific structures in the prediction models, allowing to compare the importance of explanatory variables between different models[37]. This approach allows us to compare different machine learning models independently of their mathematical and statistical performance. The main idea is to assess how much the performance of a model changes when a given variable is neutralized (or controlled). Thus, if an independent variable is important for predicting the dependent variable, the performance of the models will worsen proportionally. For this method, we used 50 resamples with the training data to obtain the average missing AUC of the total of 50 resamples for each variable ($n = 2000$).

## Sensitivity analysis

To assess the consistency of the importance of the variables, we performed a sensitivity analysis and trained a random forest model for each country group, the results are presented in Figures S1–S6 and Tables S3–S5 in the supplementary material.

## Reporting summary

Further information on research design is available in the Nature Portfolio Reporting Summary linked to this article.

Article

## Data availability

The study data are available with restricted access due to their large size and format, and can be accessed by emailing the corresponding author. Further details of the homogenized data generated in this study are provided in the Supplementary Information/Source Data file. Source data are provided with this paper. The data for the figures and tables generated in this study are provided in the Supplementary Information/Data Source file. The data used in this study are available in the database contained in the repository referenced in the code availability statement. All third-party datasets used in this study are publicly available and openly accessible for research purposes. As these data are already in the public domain, no additional permissions or approvals were required for their use. All corresponding data sources have been fully cited in the manuscript and can be accessed by the general public according to the terms and conditions established by the data providers. Source data are provided with this paper.

## Code availability

Computer code is available from GitHub under https://github.com/ZamudioSosaAlejandro/Decoding-health-disparities-by-gender-ethnicity-and-chronic-diseases-.

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

## Acknowledgements

Financial support for this study was provided by Medical Research Council (MRC: Grant Number: MC_PC_MR/T023678/1). The authors declare no conflicts of interest. All authors have approved the paper for submission. I also tell you that the manuscript is not being considered in another journal.

## Author contributions

C.C., P.H., A.L.M., and D.R. contributed to the study conception and design. Material preparation and data collection was performed by A.Z.S., D.M.C., J.A.O., and C.A.; Analysis were performed by A.Z.S. The first draft of the manuscript was written by C.C. and A.Z.S. All authors commented on previous versions of the manuscript. All authors read and approved the final manuscript.

## Competing interests

The authors declare no competing interests.
