## [Transparent Peer Review file · Nature Communications]

Decoding health disparities: gender, ethnicity, and chronic diseases in three countries of Latin Americans

Corresponding Author: Mr Carlos Chivardi

Version 0:

Reviewer comments:

Reviewer #1

(Remarks to the Author)

1. Abstract: The reported number of observations appears misleading, as it does not reflect the actual number of observations collected in the surveys. Additionally, while exploring the characteristics of individuals with a specific diagnosis is important, it does not adequately address the broader process of social determinants.
2. There is a substantial body of literature analyzing health inequalities in Latin America that consider the roles of gender and ethnicity. I recommend reviewing the following Google Scholar search for relevant papers: Health Inequalities in Latin America.
3. The aim of the paper is not entirely clear. It seems to focus on describing the profile of the diagnosed population rather than exploring the connections between social determinants and chronic diseases. Please clarify your primary objective.
4. The Methods section should specify whether the questions and measurements across the three data sources are comparable and detail how the data was harmonized. Which specific questions from each survey were utilized in your analysis?
5. Since the indicator is based on previous diagnoses, it may reflect disparities in access to health services influenced by social determinants. How have you accounted for this in your analysis? Please explain the rationale behind the groups formed for the analyses.
6. It is misleading to report 97 million observations when none of the surveys account for that number of individuals. It seems likely this figure is derived from survey weights. If this is the case for your sample, could you clarify the actual number of observations included in your analysis?
7. The Mexican ENSANUT defines adults as individuals aged 20 years and older, which raises questions about the inclusion of those aged 18 and 19. Additionally, it is difficult to verify the data used as there are no references to the specific surveys. The ENSANUT 2018 dataset from Ecuador (available online at Ecuador Encifras) appears not to include variables related to chronic diseases. In Mexico, the proportion of adults with chronic conditions appears to be lower than expected.
8. Your analysis appears to focus on exploring correlations rather than assessing the influence of social determinants. Please clarify this distinction. It seems your analysis predicts diagnoses rather than examining the conditions themselves.
9. The Results section should incorporate measures of uncertainty. It is unclear what p-values are reported in Table 1—specifically, which comparisons are being made.
10. Given that chronic conditions tend to evolve with age, significant age differences among the populations studied may contribute to variations in chronic condition prevalence. Older populations are generally expected to demonstrate higher prevalence rates.
11. The Discussion section reiterates commonly known facts, such as the economic disparities between indigenous and non-indigenous populations. It would benefit from deeper insights or new perspectives on these issues.

Reviewer #2

(Remarks to the Author)

The study is interesting, however, the current version requires a series of modifications:

- a) In the introduction, it is necessary to add the theoretical justification of all the independent variables with the dependent ones, not only those of gender and race or ethnicity
- b) In the introduction, add the gaps in the literature that your study covers because the relationships sought to be analyzed

are widely reported in the literature of both the countries under study and in Latin America.

c)The study is not from Latin America, but from three countries, so it is necessary to compose both the title and the objective.

d)The objective is very ambiguous, it is necessary to clarify it

e) The authors never clarify why their study was carried out from 2018 to 2019, a pre-pandemic period when several studies have shown that COVID-19 introduced changes (positive and negative) in several aspects of public health. Is necessary justify this decision beyond the availability of data in the two countries of the study

f) In the case of the databases of Ecuador and Brazil, the latest versions of the databases are 2018 and 2019 respectively, but in the case of Mexico, there are more current versions of the ENSANUT, so why not use it? Above all, because the study as it is currently already uses databases from different years

g) In the methodology, formalize the statistical models.

h) One of the concerns I have is that methodologically the three surveys used have different data collection methodologies, so I suggest that a table be prepared where the main methodological aspects in the data collection are explained and in this way, the differences are identified, which can then be reported as limitations of the study. This is especially because the results are being reported globally.

i)Due to the population characteristics (gender, ethnicity, etc.), it is necessary to add the behavior of the variables for each country to Table 1.

j)Likewise, for figures 1 to 4, it is necessary to do them by country as well, because the social conditions in each one are different. This will also provide more information to improve the analysis and discussion of your data.

k) In the discussion, it is necessary to focus on those novel aspects of your study, because, in each of the countries you are investigating, there are already various investigations and reports from those who generated the surveys, on how gender, race, or ethnicity, as well as home conditions, influence chronic degenerative diseases.

l) Include the limitations of the study in the conclusions.

Reviewer #3

(Remarks to the Author)

This study analyzed health survey data from adults in Brazil, Ecuador and Mexico (2018-2019) to investigate how gender and ethnicity influence chronic disease outcomes through social determinants. Using random forest models, the paper reports that education was a stronger predictor of chronic diseases for women (removing it reduced model performance by 59.6%), while occupation was more influential for men (31.6% reduction). The models performed best at predicting chronic diseases among indigenous and black populations, suggesting social determinants have particularly strong impacts in these groups. Importantly, the study revealed that education levels had opposite effects by gender: higher education was associated with increased disease risk in men but decreased risk in women. Basic services like water and sewage access generally reduced disease risk, except among indigenous populations where the relationship was more complex. Indigenous people faced the worst socioeconomic conditions overall, while mixed-ethnicity individuals had the best. The findings highlight the need for targeted public health interventions that account for both gender and ethnic differences, such as focusing on educational access for women and employment opportunities for men, particularly in indigenous and black communities.

The paper makes an important contribution to our current understanding of inequality in health outcomes in Latin America. However, I have several suggestions regarding the motivation of the approach taken, sample selection and the methodological approach that could strengthen the paper further.

Main comments:

1. The paper needs a stronger motivation for the selection of these three countries. Even with the focus on Latin America there are tons of other data sources like DHS, MICS, STEPS and country health surveys (e.g., Argentina (ENNyS2), Chile (ENS), Colombia (ENDS) and Uruguay).
2. The authors should justify pooling the observations of three countries that differ so much in the cultural underpinning of gender and ethnicity as well as health systems, social protection networks, economic and social development. (How much of the variation is explained by country fixed effects?)
3. Could the study be enriched by looking at information on STEPS or some other health surveys where there is self-reported and actual tests are performed (e.g., through blood tests or blood pressure tests).
4. While the paper offers a theoretical framing of gender as a socially constructed set of roles, relationships, and power dynamics, it fails to explain how this complex concept was operationalized in the empirical analysis. Despite using data from three different national health surveys in Brazil, Ecuador, and Mexico, the paper provides no information about how gender was coded in these surveys, whether coding was consistent across countries, if gender was self-reported or assigned, or how non-binary gender identities were handled. This omission creates a significant gap between the paper's theoretical framework and its empirical implementation. Understanding the exact measurement of gender is crucial for interpreting the

findings about gender differences in health outcomes and social determinants, especially given potential cross-cultural variations in gender conceptualization across Latin America.

5. The paper's treatment of ethnic classification raises methodological concerns. While it analyzes ethnic disparities across four broad categories (mixed ethnicity, indigenous, black, and "others"), it fails to explain how these categories were constructed and harmonized across Brazil, Ecuador, and Mexico - three countries with distinct ethnic classification systems and historical contexts. The paper does not specify whether ethnicity was self-reported or assigned, nor does it justify critical methodological decisions such as combining mulatto, montuvio, and whites into an "other" category. This is particularly problematic given Latin America's complex racial and ethnic hierarchies and the varying meanings of ethnic categories across countries. Without a clear explanation of how ethnic categories were operationalized and standardized, it becomes difficult to interpret the reported ethnic disparities in health outcomes and evaluate the broader implications of the study for understanding health inequities in Latin America.

6. Are there enough observations to make reliable inference in each of the ethnicity/gender cells in each country?

7. The paper doesn't do a sufficient job explaining why machine learning methods are preferable for this analysis. The only real justification appears on page 3, lines 53-55: "Machine learning can reveal linear and non-linear relationships and often outperforms traditional models." It would be useful to compare the results with more traditional approaches.

8. The paper says that they have more than 96 million individuals. Is this because they use the expansion weight? If this is the case it will be useful to describe the statistical procedures in terms of individual survey observations without using expansion weights.

9. The use of expansion weights allows researchers to generate population-representative statistics. However, their application in machine learning contexts, particularly for measures of model fit and variable importance, raises methodological concerns. The core issue is that expansion weights, which replicate individual observations to match population totals, can distort the assumptions underlying machine learning algorithms. Random forests and similar methods assume independent observations and rely on actual data variation to assess feature importance and model fit. Using weighted observations could artificially inflate model performance metrics and bias variable importance measures by creating dependencies in the data. The authors need to address more explicitly how these survey design elements are incorporated into the analysis and clearly distinguish between using weights for population inference versus model estimation.

Minor comments:

10. Line 63, the word "Brazil" is missing.

11. Line 72, I would suggest using "employment status" rather than "occupation" to describe formal/informal employment, unemployed, or retired categories.

Version 1:

Reviewer comments:

Reviewer #1

(Remarks to the Author)

While most of my previous comments were addressed (thanks for the answers), there are some remaining concerns, that are even reinforced by the authors. The first one related to the number of observations and use of survey weights. Authors indicate that "The total sample size collected across the three national surveys is approximately 170,000 individuals for each gender/ethnic group", that is unlikely as the total number of observations is about that number, as the samples for the surveys are not expected to collect the same number of observations for the mentioned groups. It seems that authors are using weights as multipliers. A "sample" could include several identical observations if that is the case. The second is related to the analysis; while I see the advantages of the approach, it is unclear how potential interactions are handled in the analysis. Individuals with a given level of education may also be more likely to have a specific type of employment or a differential probability of access to services. As far as I understand, random forest selects only some variables as the nodes, so may not include relevant interactions in each tree.

The previous comments on the differences in NCD prevalence for Mexico and Ecuador's data remain unanswered. If I understand correctly, authors are using for Ecuador the Base 1 from the ENSANUT, with the question about a health issue in the previous 30 days. This does not seem comparable at all to the data from Mexico and Brazil.

As the analysis is by sex and not by gender, it is essential to review the background and interpret the results accordingly. It is unclear how the report from Mexico and Brazil of ever diagnosed cases can be compared with the 30-day period in the Ecuador data.

1. Still a more convincing argument is required on why those 3 countries. It seems to me that could be convenience, in the sense of previous knowledge of the datasets and about the context. Arguing that is due to harmonizable data assumes other

surveys could not as harmonizable. While that could be true, there is no evidence provided on that. Having Brazil and Mexico plus almost any other country will result in a similar share of the regional population. It would also important to recognize the real motive as a limitation.

2. There is no answer to the specific question on the variation explained by country fixed effects.

3. Not clear on this answer. At least the Mexican ENSANUT included biomarkers, so that data is already available in the surveys you are pooling.

4. Surveys ask about sex not gender. This should be explicit and discussed how that limits the analysis related to gender.

5. Still not clear about the ethnic classification; how individuals from Mexico were coded? As not included in the survey in Mexico, the authors are assuming no blacks in Mexico? It would be useful to have a table with the descriptive data by country. Assuming census data, it could be well the case that you have mostly all blacks in Brazil, the majority of indigenous from Ecuador, and the majority of not sure if mixed or others from Mexico, depending on how non-indigenous from Mexico were coded.

6. Sampling weights were used to multiply observations? That seems not to be appropriate. Your statistical power is an artifact; you are treating a clustered sample as a simple random sample of an artificially homogenous population.

7. Either you are using weights to create additional observations artificially or not using them at all . Your methods sections suggest that you random sample of 170k per group is from the 96 million individuals that result from creating new observations using the weights as multipliers. If this is the case, there is an issue here as observations are not independent at all.

Reviewer #2

(Remarks to the Author)

The paper met the suggested recommendations, so that the methodology and results are now more transparent, showing trends in the study countries. These findings can inform future research and demonstrate public policy implications.

Version 2:

Reviewer comments:

Reviewer #1

(Remarks to the Author)

All my previous comments have been addressed, thanks.

Manuscript Title: Decoding health disparities: gender, ethnicity, & chronic diseases in Latin Americans with individual data & machine learning

Manuscript ID: NCOMMS-24-63370-T

Comment	Response	Changes to the manuscript
Reviewer #1		
1. Abstract: The reported number of observations appears misleading, as it does not reflect the actual number of observations collected in the surveys. Additionally, while exploring the characteristics of individuals with a specific diagnosis is important, it does not adequately address the broader process of social determinants.	We appreciate the reviewer's comment regarding the number of observations. The reported figure of 96,726,891 represents a population-weighted estimate rather than the actual number of survey respondents. The total sample size collected across the three national surveys is approximately 170,000 individuals for each gender/ethnic group. We have revised the abstract to explicitly clarify this distinction and prevent potential misinterpretation. Regarding the concern about our focus on individuals with a specific diagnosis, we acknowledge the importance of analysing the broader process of social determinants. Our approach goes beyond merely describing affected individuals; instead, it examines how social determinants influence chronic disease diagnoses across gender and ethnic groups. By employing machine learning models, we identify predictive relationships between socioeconomic factors and disease risk, which provides insights into systemic disparities. To enhance clarity, we have refined the abstract to reflect this aspect of our study.	Revised Abstract: "In this study, we analysed data from national health surveys in Brazil, Mexico, and Ecuador (2018–2019), representing a total weighted population of 96,726,891 adults. To optimize computational efficiency, we randomly sampled 170,000 individuals from each gender/ethnic group for the analysis."
2. There is a substantial body of literature analysing health inequalities in Latin America that consider the roles of gender and ethnicity. I recommend reviewing the following Google Scholar search for relevant papers: Health Inequalities in Latin America.	Thank you for highlighting the extensive literature on health inequalities in Latin America, particularly regarding gender and ethnicity. We acknowledge the significant body of work in this area and have incorporated key references that discuss these disparities.	Revised Introduction: "Gender has emerged as a central element in understanding health disparities, defined as "the set of socially constructed roles and relationships, personality traits, attitudes, behaviours, values, relative power and influence that society differentially attributes to the two genders". Gender influences health policies, biomedical and contraceptive

Comment	Response	Changes to the manuscript
	However, our study contributes to this literature by introducing a novel methodological approach. Unlike traditional epidemiological studies that primarily use regression-based approaches, we employ machine learning techniques—specifically, random forest models—to analyze the predictive power of social determinants across gender and ethnic groups. This allows us to capture both linear and non-linear relationships in health disparities, offering a more nuanced perspective on the interplay of multiple factors. Additionally, while many studies focus on individual determinants in isolation, our approach evaluates a comprehensive set of factors—including education, occupation, infrastructure access— assessing their differential impacts across demographic groups. This intersectional analysis, particularly using large-scale data from three countries, provides new insights into how structural inequities shape chronic disease risk. In response to your suggestion, we have reviewed additional relevant studies from Google Scholar and incorporated key references where applicable to further contextualize our findings within the broader literature.	technologies, and health system responses. These disparities extend beyond biological differences, encompassing social, cultural, behavioural, and biological factors. Social scientists note that biological differences alone cannot explain health outcomes; social determinants and gender play a crucial role. Gender-based health disparities often relate to behaviours associated with gender roles and it has been found that such roles can strongly influence different health outcomes in Latin America. Men typically exhibit risk behaviours, such as increased cigarette and alcohol consumption, while women more frequently use health services and monitor physiological changes. For instance, women are more likely to seek primary medical care for illness symptoms. However, gender differences in chronic disease morbidity are sensitive to sample characteristics and analysis methods and vary by country due to cultural and social factors. Differences in health outcomes are linked to power, autonomy, poverty, and marginalization. On the other hand, there is a wide literature that indicates that in addition to gender, ethnicity has also been related to various health outcomes (infant mortality, child nutrition, maternal health, reproductive health, rheumatic diseases, etc.) that differentially affect ethnic groups in Latin America, with people of indigenous descent being the most vulnerable than any other ethnic group. In examining the gender-ethnic intersection, social determinants show differentiated roles. Szanton et al. found income and education increased morbidity and mortality risk in older women, regardless of race. Assari et al. reported that education lowered health problems for whites but not blacks, and income protected women more than men. Reducing early life adversities decreased cardiovascular disparities more among men of different ethnicities than among

Comment	Response	Changes to the manuscript
		women. Social inequalities like socioeconomic status have differential effects on health at the gender-ethnic intersection. However, although gender and ethnicity may play important roles as social determinants, they are not the only ones. Social conditions such as lifestyles, living and working situations, neighbourhood characteristics, poverty, environmental pollution, income, education and occupation have been widely studied as social determinants of different chronic diseases. In this sense, income, education and occupation characteristics have been shown to be the most important social determinants, followed closely by inequalities associated with ethnicity. Despite this, the meaning and direction of social determinants such as education, occupation and living conditions are consistent even without taking into account population and gender; people with a lower level of education, informal jobs or no work, and those lacking basic services are significantly more likely to suffer from chronic diseases and have worse treatment expectations. According to Cockerham, the above-mentioned variables have indirect effects on chronic diseases by influencing lifestyles, health behaviours, food security, stress levels as well as biological outcomes through genetic expression and generational inheritance. Health researchers emphasize the intersection of race/ethnicity, class, income, education, age, and geography in addressing health inequalities. In developed countries, black individuals and women have higher chronic disease rates than their white counterparts. In Latin America, darker-skinned individuals, especially black and indigenous women, have worse health outcomes. Despite extensive research on gender and ethnicity, there is limited intersectional analysis in Latin America, particularly

Comment	Response	Changes to the manuscript
		regarding multiple chronic diseases. Furthermore, although the relationship between social determinants and the most prevalent chronic diseases in Latin American countries has been widely studied, there is little evidence of possible non-linear relationships and differences in these relationships between ethnic and gender groups. Few studies use machine learning models to explore social determinants of chronic diseases by gender and ethnicity in Latin America. Machine learning can reveal linear and non-linear relationships and often outperforms traditional models. Our study aims to evaluate the predictive role of social determinants by gender and ethnicity in the diagnosis of chronic diseases (diabetes, cardiovascular diseases, kidney diseases, cerebrovascular diseases and obesity), which are among the leading causes of mortality in Latin America.” (Please see page 2-3, lines 30 to 76, of the main manuscript).
3. The aim of the paper is not entirely clear. It seems to focus on describing the profile of the diagnosed population rather than exploring the connections between social determinants and chronic diseases. Please clarify your primary objective.	We appreciate the reviewer’s feedback regarding the study’s aim. Our primary objective is not merely to describe the profile of the diagnosed population, but to evaluate the predictive role of social determinants in chronic disease diagnoses across different gender and ethnic groups in Latin America. To achieve this, we employ machine learning techniques (random forest models) to assess how factors such as education, occupation, and access to essential services influence chronic disease risk. By analysing the importance of these variables in predicting diagnoses—and quantifying how removing each factor impacts model performance—we provide insights into the structural drivers of health inequalities, rather than just characterizing affected individuals.	Revised aim: “Our study aims to evaluate the predictive role of social determinants by gender and ethnicity in the diagnosis of chronic diseases (diabetes, cardiovascular diseases, kidney diseases, cerebrovascular diseases and obesity), which are among the leading causes of mortality in three (Brazil, Ecuador, and Mexico) countries of Latin America.” (Please see page 3, lines 73 to 76, of the main manuscript).

Comment	Response	Changes to the manuscript
	To enhance clarity, we have revised the introduction to explicitly state this objective and ensure that the analytical focus is evident throughout the manuscript.	
4. The Methods section should specify whether the questions and measurements across the three data sources are comparable and detail how the data was harmonized. Which specific questions from each survey were utilized in your analysis?	Thank you for your valuable suggestion. We recognize the importance of ensuring comparability between data sources and have taken several steps to harmonize the data sets from Mexico (ENSANUT 2018), Brazil (PNS 2019) and Ecuador (ENSANUT 2018). Comparability and harmonization process:  • All three surveys collected self-reported health data, including diagnoses of diabetes, cardiovascular disease, kidney disease, cerebrovascular disease, and obesity. In Brazil and Mexico, respondents were explicitly asked whether a health professional had diagnosed the individual with these conditions, while in Ecuador, the question referred the question referred to whether the respondent had suffered from any of these conditions. Given the nature of chronic disease diagnoses, we assumed that individuals reporting a condition were informed of it by a health professional. • Key socioeconomic determinants (education, occupation, and access to essential services) were also measured using similar categories across surveys. • To ensure consistency, we harmonized response categories for variables such as educational attainment, employment status, and access to infrastructure. Differences in classification were standardized based on the most used categories. • A detailed mapping of variable definitions across surveys is now included in the Appendix for increased transparency. To strengthen methodological clarity, we have updated the Methods section to explicitly describe the harmonization process and included the full set of	Revised Method section: “To ensure consistency across the three national health surveys (ENSANUT 2018 in Mexico, PNS 2019 in Brazil, and ENSANUT 2018 in Ecuador), we standardized key variables related to chronic disease diagnosis and social determinants. For Mexico and Brazil, self-reported diagnostic questions were directly comparable, as respondents were uniformly asked whether a health professional had diagnosed or informed them that they had any of the chronic diseases listed: diabetes, cardiovascular disease, kidney disease, cerebrovascular disease, and obesity. For Ecuador, the question whether the person had a chronic disease and cardiovascular problems in the last 30 days was taken into account (we assumed the fact that anyone who reported having suffered from a chronic disease had to receive a diagnosis from a health professional to have this knowledge). Meanwhile, socioeconomic determinants (education, occupation, and access to essential services) were measured using equivalent categories across all data sets. Any discrepancies in classification were resolved by aligning responses with the most commonly used categories. The questions used to measure diagnosed diseases can be found in Table A1 in Appendix.” (Please see page 5, lines 108 to 119, of the main manuscript).

Comment	Response	Changes to the manuscript
	survey questions in the Supplementary materials (Table S1).	
5. Since the indicator is based on previous diagnoses, it may reflect disparities in access to health services influenced by social determinants. How have you accounted for this in your analysis? Please explain the rationale behind the groups formed for the analyses.	We appreciate this insightful comment regarding the potential influence of healthcare access disparities on self-reported diagnoses. We acknowledge that variations in access to health services—driven by social determinants—could affect the likelihood of receiving a formal diagnosis, particularly among marginalized populations. Addressing Healthcare Access Disparities: To account for this, we incorporated infrastructure-related variables like piped water, sewage, garbage collection, and urban/rural location, all of these factors help capture differences in living conditions that may correlate with disparities in health service availability and utilization. Additionally, our machine learning approach allowed us to assess the relative importance of different social determinants in predicting disease diagnoses, highlighting how education, employment, and infrastructure interact with health outcomes. Rationale for Group Formation: Our decision to stratify the analysis by gender and ethnicity was based on well-documented evidence that these factors significantly shape health inequalities in Latin America and the world (1–20). Prior research has shown that ethnic minorities (especially indigenous and black populations) face systemic barriers to healthcare, and that gender further compounds these disparities. By analysing eight distinct gender-ethnicity subgroups, we aimed to capture intersectional disparities and evaluate whether social determinants impact health outcomes differently across populations.	

Comment	Response	Changes to the manuscript
	To enhance clarity, we have revised the Methods section to explicitly address these considerations and the rationale behind the group classifications.	
6. It is misleading to report 97 million observations when none of the surveys account for that number of individuals. It seems likely this figure is derived from survey weights. If this is the case for your sample, could you clarify the actual number of observations included in your analysis?	We appreciate the reviewer’s concern regarding the reported number of observations. To clarify, the 97 million figure represents the weighted population estimate, derived using survey expansion factors to provide a representative count of adults (20+ years) in Mexico, Ecuador, and Brazil. However, the actual number of survey respondents included in our analysis was 42,068 for Mexico, 96,111 for Ecuador, and 40,088 for Brazil for Brazil. In response to this comment, we have revised the Methods section to explicitly distinguish between the actual number of observations and the weighted population estimates used for inference. This ensures greater transparency and avoids potential misinterpretation.	“To estimate the total adult population (20+ years) in each country, we applied survey expansion factors, yielding a weighted population estimate of approximately 97 million individuals. These weights allow the sample to reflect the broader population distribution within each country. However, the actual number of survey respondents included in our analysis—before applying expansion factors—was 42,068 for Mexico, 96,111 for Ecuador, and 40,088 for Brazil. This distinction is important to ensure clarity between the weighted estimates used for inference and the raw number of observations collected.” (Please see page 4, lines 82 to 88, of the main manuscript).
7. The Mexican ENSANUT defines adults as individuals aged 20 years and older, which raises questions about the inclusion of those aged 18 and 19. Additionally, it is difficult to verify the data used as there are no references to the specific surveys. The ENSANUT 2018 dataset from Ecuador (available online at Ecuador Encifras) appears not to include variables related to chronic diseases. In Mexico, the proportion of adults with chronic conditions appears to be lower than expected.	Thank you for your comments on the definitions of adulthood and data sources. We recognize the importance of ensuring transparency and methodological rigor in our study. While Mexico's ENSANUT 2018 officially defines adults as individuals aged 20 years or older, the Ecuador and Brazil surveys include individuals aged 18 and 19 years (ENSANUT Ecuador 2018 and PNS Brazil 2019). To increase data consistency across the three countries, we decided to restrict our analysis to individuals aged 20 years or older, excluding records for those aged 18 and 19. We reran the analysis with this adjustment and found that the results changed very little, with the importance of the variables remained consistent. The Results section was updated accordingly.	The data were obtained from: Mexico: Encuesta Nacional de Salud y Nutrición (ENSANUT) 2018. Ecuador: Encuesta Nacional de Salud y Nutrición (ENSANUT) 2018. Brazil: Pesquisa Nacional de Saúde (PNS) 2019. (Please see page 4, lines 80 to 82, of the main manuscript).

Comment	Response	Changes to the manuscript
	We recognize the importance of clearly referencing data sources. In response to this comment, we have explicitly cited the official sources for each data set in the Methods section.	
8. Your analysis appears to focus on exploring correlations rather than assessing the influence of social determinants. Please clarify this distinction. It seems your analysis predicts diagnoses rather than examining the conditions themselves.	We appreciate the reviewer’s comment regarding the nature of our analysis. Our study does not focus solely on correlations but rather employs machine learning models (random forest) to assess the predictive power of social determinants in diagnosing chronic diseases. Unlike traditional correlation-based approaches, our analysis evaluates the relative importance of different social determinants (e.g., education, occupation, infrastructure access) in predicting chronic disease risk across gender and ethnic groups. While our models predict chronic disease diagnoses based on social determinants, they also highlight which factors have the strongest association with health disparities. The use of variable importance measures and AUC loss analyses allows us to infer the extent to which different determinants contribute to disparities. We acknowledge that our approach does not establish causal relationships, but it provides actionable insights for policymakers by identifying key drivers of chronic disease risk in marginalized populations. Also, numerous studies in health economics have already used similar approaches (21).	
9. The Results section should incorporate measures of uncertainty. It is unclear what p-values are reported in Table 1—specifically, which comparisons are being made.	We appreciate the reviewer’s feedback regarding the need for measures of uncertainty in the Results section. To enhance clarity, we have revised the Table 1 caption and Methods section to explicitly describe these statistical comparisons. Additionally, the	We have incorporated in the Results section “The chi-square test was used for categorical variables, and Kolmogorov–Smirnov for age distribution.”

Comment	Response	Changes to the manuscript
	machine learning methods we have used do not have p-values or uncertainty measures since hypothesis testing is not performed.	(Please see page 7, line 168 to 169, of the main manuscript).
10. Given that chronic conditions tend to evolve with age, significant age differences among the populations studied may contribute to variations in chronic condition prevalence. Older populations are generally expected to demonstrate higher prevalence rates.	Thank you for your insightful comment regarding the role of age in chronic disease prevalence. We fully acknowledge that age is a key determinant of chronic disease risk, and differences in age distributions among the studied populations could influence our findings. To address this concern, we included age as a control variable in our machine learning models, ensuring that the effect of social determinants was assessed independently of age-related variations. Unlike traditional regression-based approaches that rely on p-values for significance testing, machine learning methods, such as random forest models, assess variable importance through techniques like Mean Decrease in Impurity (MDI; (22)) and SHAP values (23). For our study, we used an agnostic approach that does not assume specific structures in the algorithms. These methods allow us to evaluate the relative contribution of age in predicting chronic disease diagnoses while accounting for interactions with other social determinants. In this sense, in the partial dependence graphs (24) we show the change in probability of chronic disease by controlling for other predictor variables, including age. This ensures that relationship between social determinants and disease risk is evaluated while accounting for age-related effects. To enhance clarity, we have revised the Methods section to explicitly describe how age was incorporated into the model and how its influence was	In response to this comment, we have revised the Methods and Results sections to explicitly mention the inclusion of age as a control variable and to clarify how age differences were addressed in our analysis. “Additionally, given the important predictive role of age (as a continuous variable) and country (as a categorical variable) in the diagnosis of chronic diseases, we included age and country as control variables in the models. Since we focused on social determinants, these control variables were not included in the results of the importance of the variables.” (Please see page 4, lines 99 to 102, of the main manuscript).

Comment	Response	Changes to the manuscript
	assessed through feature importance and partial dependence analyses.	
11. The Discussion section reiterates commonly known facts, such as the economic disparities between indigenous and non-indigenous populations. It would benefit from deeper insights or new perspectives on these issues.	We appreciate the reviewer’s feedback on the Discussion section and recognize the importance of providing deeper insights beyond commonly known disparities. While we acknowledge well-established economic inequalities between Indigenous and non-indigenous populations. Specifically, we highlight how our machine learning approach allows for a more nuanced understanding of the relative importance of different social determinants in predicting chronic disease diagnoses, moving beyond traditional regression-based analyses. Additionally, we contextualize our findings within broader structural inequities, discussing how specific policies and healthcare access barriers contribute to the observed disparities. To address this, we have revised the Discussion section to move beyond broad socioeconomic disparities and emphasize these unique findings, their policy implications, and the potential mechanisms driving these relationships.	“Our study highlights the importance of analysing social determinants as an interconnected system rather than in isolation, as demonstrated by the strong predictive performance of our machine learning models. A key finding is the differential role of education and employment across genders, with education being more critical for women—likely due to its impact on health literacy and preventive care—while employment status plays a stronger role for men, potentially due to economic stability and healthcare access. This gendered effect, which has been poorly documented in the literature, underscores the need for targeted health policies that prioritize education for women and employment-related interventions for men. Additionally, our results reveal that social determinants have a stronger and more complex influence among indigenous and black populations, with interactions between multiple factors playing a more significant role compared to mixed and other ethnic groups. Notably, higher education was paradoxically associated with an increased likelihood of chronic disease diagnosis among indigenous individuals, possibly reflecting improved healthcare access leading to higher detection rates rather than increased disease burden. Another possible explanation is that higher education provided this population with greater purchasing power and this in turn reflects changes in health behaviours in nutrition and substance use. Furthermore, infrastructure access (piped water, sewage, and garbage collection) emerged as a significant yet unevenly distributed determinant, disproportionately affecting indigenous women. Importantly, much of the research on social determinants of health (SDH) focuses not only on identifying whether specific determinants exert an

Comment	Response	Changes to the manuscript
		effect, but also on quantifying the magnitude of these effects. Traditional statistical models often require strict assumptions about linearity and variable independence, which may obscure the true magnitude of influence, particularly when determinants interact in complex, non-linear ways. In contrast, machine learning models excel in capturing these intricate relationships without imposing rigid model structures. This flexibility allows for more accurate and nuanced estimation of effect sizes across diverse population subgroups, as demonstrated by the strong and interpretable patterns observed in our analyses. By leveraging machine learning, our study moves beyond binary associations to provide a clearer picture of how multiple, interrelated SDH factors jointly shape health outcomes. These findings emphasize the need for comprehensive public health strategies that address both social and environmental determinants. Policymakers should implement education-based interventions for women, employment-focused policies for men, and holistic infrastructure improvements for indigenous and black communities to mitigate structural inequalities and improve health outcomes.” (Please see page 16-17, lines 339 to 362, of the main manuscript).
Reviewer #2		
a)In the introduction, it is necessary to add the theoretical justification of all the independent variables with the dependent ones, not only those of gender and race or ethnicity	Thank you for your valuable feedback. We acknowledge the importance of providing a theoretical justification for all independent variables in relation to the dependent variable, beyond gender and ethnicity. In response, we have revised the Introduction section to explicitly explain the rationale behind including education, employment status, and infrastructure access (piped water, sewage, garbage collection) as key determinants of chronic disease risk.	“However, although gender and ethnicity may play important roles as social determinants, they are not the only ones. Social conditions such as lifestyles, living and working situations, neighbourhood characteristics, poverty, environmental pollution, income, education and occupation have been widely studied as social determinants of different chronic diseases. In this sense, income, education and occupation characteristics have been shown to be the

Comment	Response	Changes to the manuscript
		most important social determinants, followed closely by inequalities associated with ethnicity. Despite this, the meaning and direction of social determinants such as education, occupation and living conditions are consistent even without taking into account population and gender; people with a lower level of education, informal jobs or no jobs, and those lacking basic services are significantly more likely to suffer from chronic diseases and have worse treatment expectations. According to Cockerham¹⁴, the above-mentioned variables have indirect effects on chronic diseases by influencing lifestyles, health behaviors, food security, stress levels as well as biological outcomes through genetic expression and generational inheritance.” (Please see page 2, lines 52 to 63, of the main manuscript).
b) In the introduction, add the gaps in the literature that your study covers because the relationships sought to be analyzed are widely reported in the literature of both the countries under study and in Latin America.	Thank you for this insightful comment. We acknowledge that the relationships between social determinants and chronic diseases have been widely studied in both the countries under analysis and in Latin America. However, our study addresses several key gaps in the existing literature by:  1. Using machine learning models to assess the predictive power of social determinants, rather than traditional regression-based approaches. This allows for a more comprehensive and accurate estimation of the effects and nonlinear exploration of interactions between determinants, which is often overlooked in previous studies. 2. Investigating the differential role of education and employment across genders, revealing that education is more influential for women, while employment is more 	Introduction section: “Despite extensive research on gender and ethnicity, there is limited intersectional analysis in Latin America, particularly regarding multiple chronic diseases. Furthermore, although the relationship between social determinants and the most prevalent chronic diseases in Latin American countries has been widely studied, there is little evidence of possible non-linear relationships and differences in these relationships between ethnic and gender groups. Few studies use machine learning models to explore social determinants of chronic diseases by gender and ethnicity in Latin America. Machine learning can reveal linear and non-linear relationships and often outperforms traditional models”

Comment	Response	Changes to the manuscript
	critical for men—a finding that has been poorly documented in the region. 3. Examining the complex interactions of social determinants in indigenous and black populations, where we found that the combined effect of social determinants is stronger than in mixed or other ethnic groups. This highlights the need for intersectional policy approaches that go beyond broad generalizations about health disparities. 4. Providing new insights into the association between education and chronic disease risk in indigenous populations, where we observed a paradoxical trend: higher education levels were associated with an increased likelihood of chronic disease diagnosis—possibly due to improved healthcare access and diagnosis rates, rather than an actual increase in disease prevalence. In response to this comment, we have revised the introduction section to explicitly outline these gaps in the literature and how our study adds new perspectives to the discussion of health disparities in Latin America. Additionally, we have explicitly added the gaps we addressed in the discussion section.	(Please see page 3, lines 67 to 73, of the main manuscript). Discussion section: “While the relationships between social determinants and chronic diseases have been widely studied in Latin America, significant gaps remain in the literature. First, most previous studies have relied on traditional regression-based approaches, which may not fully capture the complex, nonlinear interactions between social determinants. Second, social determinants are well-studied, few studies in the region have explored their differential impact across genders. Third, there is limited research on how social determinants interact specifically within indigenous and black populations, despite these groups facing some of the greatest health inequities. By addressing these gaps, our study contributes a novel perspective on the role of social determinants in health disparities across Latin America and informs targeted policy interventions to reduce these inequities.” (Please see page 13, lines 255 to 262, of the main manuscript).
c)The study is not from Latin America, but from three countries, so it is necessary to compose both the title and the objective.	We appreciate the reviewer’s comment regarding the scope of our study. While our analysis focuses on three countries—Brazil, Ecuador, and Mexico—rather than all of Latin America, these countries collectively represent a significant proportion of the region’s population. Specifically, Brazil, Mexico, and Ecuador together account for over 50% of the total Latin American population, making them highly relevant for studying social determinants of health in the region.	The title was modified: “Decoding health disparities: gender, ethnicity, and chronic diseases in three countries of Latin Americans with individual data & machine learning” The objective was modified: “Our study aims to evaluate the predictive role of social determinants by gender and ethnicity in the diagnosis of chronic diseases (diabetes, cardiovascular

Comment	Response	Changes to the manuscript
	Given this, we recognize the need for greater precision in how we describe the study’s scope. In response, we have revised both the title and the objective statement to clarify that our findings are based on these three countries while maintaining relevance to broader health disparities in Latin America.	diseases, kidney diseases, cerebrovascular diseases and obesity), which are among the leading causes of mortality in three (Brazil, Ecuador, and Mexico) countries of Latin America.” (Please see page 3, lines 73 to 76, of the main manuscript).
d)The objective is very ambiguous, it is necessary to clarify it	Thank you for your feedback regarding the clarity of the study’s objective. We acknowledge the need for greater precision in articulating the research aim. In response, we have revised the objective statement to clearly reflect the study's focus on assessing the predictive role of social determinants (education, employment status, and infrastructure access) in chronic disease diagnoses across different gender and ethnic groups in Brazil, Ecuador, and Mexico. Unlike previous studies that primarily describe health disparities, our research utilizes machine learning models to evaluate the relative importance and interactions of these determinants, offering new insights into how social and structural inequalities shape chronic disease risk.	“Our study aims to evaluate the predictive role of social determinants by gender and ethnicity in the diagnosis of chronic diseases (diabetes, cardiovascular diseases, kidney diseases, cerebrovascular diseases and obesity), which are among the leading causes of mortality in three (Brazil, Ecuador, and Mexico) countries of Latin America.” (Please see page 3, lines 73 to 76, of the main manuscript).
e) The authors never clarify why their study was carried out from 2018 to 2019, a pre-pandemic period when several studies have shown that COVID-19 introduced changes (positive and negative) in several aspects of public health. Is necessary justify this decision beyond the availability of data in the two countries of the study	We appreciate the reviewer’s comment regarding the timeframe of our study. The decision to use 2018–2019 data was primarily driven by the need to examine pre-pandemic health disparities, allowing for an analysis unaffected by the substantial disruptions caused by COVID-19. While the pandemic had significant effects on healthcare access, employment, and social conditions, its impact was highly heterogeneous, varying across countries and demographic groups based on government responses and public health policies (25). By focusing on pre-pandemic data, our study captures	

Comment	Response	Changes to the manuscript
	a baseline understanding of the structural social determinants influencing chronic disease risk, without the confounding effects of emergency public health measures, economic fluctuations, or healthcare system disruptions introduced by COVID-19 (26). To enhance clarity, we have revised the Methods section to explicitly justify this decision beyond data availability, reinforcing the importance of capturing structural determinants before pandemic-driven shifts in public health and socioeconomic conditions.	
f) In the case of the databases of Ecuador and Brazil, the latest versions of the databases are 2018 and 2019 respectively, but in the case of Mexico, there are more current versions of the ENSANUT, so why not use it? Above all, because the study as it is currently already uses databases from different years	Thank you for your valuable observation about the years considered in our study. Although it is true that in the case of Mexico, there are more updated versions of health surveys (e.g. ENSANUT 2022), our decision was based on selecting the surveys available in the three countries with the shortest distance between them, that is, we selected the available surveys from the three countries that were closest to each other, this mainly based on the evidence that has shown that social phenomena can affect the relationships between social determinants and diseases (27)	
g) In the methodology, formalize the statistical models.	Thank you very much for your suggestion to formalize the random forest model in the article. In this sense, we have added in the method section a mathematical formulation of the random forest model and a general explanation of how it works for classification tasks.	
h) One of the concerns I have is that methodologically the three surveys used have different data collection methodologies, so I suggest that a table be prepared where the main methodological aspects in the data collection are explained and, in this way, the differences are identified, which can then be reported as limitations of the study. This is especially because the results are being reported globally.	Thank you for this valuable suggestion. We acknowledge that the three surveys used in our study (ENSANUT Mexico 2018, ENSANUT Ecuador 2018, and PNS Brazil 2019) have different data collection methodologies, which could introduce potential comparability challenges. To address this concern, we have now included a methodological comparison table in the Supplementary Materials, summarizing key	“One key limitation of this study is the methodological differences between the three national health surveys used (ENSANUT Mexico 2018, ENSANUT Ecuador 2018, and PNS Brazil 2019). Each survey employed distinct sampling strategies, data collection methods, and weighting procedures, which may introduce variability in the comparability of the results. While we harmonized key variables (e.g., education,

Comment	Response	Changes to the manuscript
	aspects of each survey, including sampling design, data collection methods, weighting procedures, and relevant variable definitions. This information helps identify potential differences in survey methodologies and ensures transparency regarding how these factors may influence our findings. Additionally, we have updated the Limitations section to explicitly discuss how these methodological variations could affect the interpretation of results. While our study applies harmonization techniques to align variables across surveys, we recognize that certain differences may persist, and we now acknowledge this as a study limitation. In addition, to address how this issue may affect the results, we have performed the predictive analyses by country and added them to the supplementary materials section (Figure S10-S15). These results show great congruence with the general results since we have also included the country as a control variable.	employment status, and chronic disease diagnoses) to ensure consistency across datasets, some differences may still persist. To enhance transparency, we have included a methodological comparison table (Supplementary Materials) outlining key differences and similarities between the surveys. “ (Please see page 15, lines 320 to 326, of the main manuscript).
i)Due to the population characteristics (gender, ethnicity, etc.), it is necessary to add the behaviour of the variables for each country to Table 1.	Thank you very much for your comments on the descriptive analyses. In order not to make the results section larger and in line with the machine learning analyses by country, we consider it pertinent to add the descriptive analyses by country in the Supplementary materials (Table S2-S4).	
j) Likewise, for figures 1 to 4, it is necessary to do them by country as well, because the social conditions in each one are different. This will also provide more information to improve the analysis and discussion of your data.	Thank you very much for the recommendation to add all the analyses but by country. We have taken this suggestion into account, and we have replicated figures 1 to 4 but now with data from each country in the Supplementary material section in Table S2-S4.	

Comment	Response	Changes to the manuscript
k) In the discussion, it is necessary to focus on those novel aspects of your study, because, in each of the countries you are investigating, there are already various investigations and reports from those who generated the surveys, on how gender, race, or ethnicity, as well as home conditions, influence chronic degenerative diseases.	Thank you for this valuable suggestion. We acknowledge that prior research in Mexico, Ecuador, and Brazil has extensively examined the influence of gender, race/ethnicity, and home conditions on chronic diseases. However, our study provides new contributions by employing machine learning models to assess the predictive power and interactions of these social determinants, rather than relying on traditional statistical approaches. Specifically, our findings reveal three key novel aspects that have been poorly documented in the existing literature:  1. The differential role of education and employment by gender—we found that education is a stronger predictor for women, likely due to its role in health literacy and preventive behaviors, while employment status is more important for men, potentially linked to financial stability and healthcare access. 2. The increased predictive strength of social determinants among indigenous and black populations, suggesting that systemic inequalities amplify the effects of education, employment, and infrastructure conditions more than in mixed and other ethnic groups. This underscores the need for intersectional policy interventions. 3. The paradoxical association between education and chronic disease diagnoses in indigenous populations, where higher education was linked to increased diagnosis rates. This may indicate improved healthcare access and detection rather than an actual increase in disease burden, a pattern that has not been widely explored in Latin America. 	“Our study highlights the importance of analysing social determinants as an interconnected system rather than in isolation, as demonstrated by the strong predictive performance of our machine learning models. A key finding is the differential role of education and employment across genders, with education being more critical for women—likely due to its impact on health literacy and preventive care—while employment status plays a stronger role for men, potentially due to economic stability and healthcare access. This gendered effect, which has been poorly documented in the literature, underscores the need for targeted health policies that prioritize education for women and employment-related interventions for men. Additionally, our results reveal that social determinants have a stronger and more complex influence among indigenous and black populations, with interactions between multiple factors playing a more significant role compared to mixed and other ethnic groups. Notably, higher education was paradoxically associated with an increased likelihood of chronic disease diagnosis among indigenous individuals, possibly reflecting improved healthcare access leading to higher detection rates rather than increased disease burden. Another possible explanation is that higher education provided this population with greater purchasing power and this in turn reflects changes in health behaviours in nutrition and substance use. Furthermore, infrastructure access (piped water, sewage, and garbage collection) emerged as a significant yet unevenly distributed determinant, disproportionately affecting indigenous women. Importantly, much of the research on social determinants of health (SDH) focuses not only on identifying whether specific determinants exert an effect, but also on quantifying the magnitude of these effects. Traditional statistical models often require strict assumptions about linearity and variable

Comment	Response	Changes to the manuscript
	In response to this comment, we have revised the Discussion section to emphasize these novel contributions and clearly distinguish our study from existing reports on health disparities in Latin America.	independence, which may obscure the true magnitude of influence, particularly when determinants interact in complex, non-linear ways. In contrast, machine learning models excel in capturing these intricate relationships without imposing rigid model structures. This flexibility allows for more accurate and nuanced estimation of effect sizes across diverse population subgroups, as demonstrated by the strong and interpretable patterns observed in our analyses. By leveraging machine learning, our study moves beyond binary associations to provide a clearer picture of how multiple, interrelated SDH factors jointly shape health outcomes” (Please see page 16, lines 339 to 362, of the main manuscript).
l) Include the limitations of the study in the conclusions.	Thank you for your suggestions. We have addressed all the limitations in the Discussion section and have incorporated the additional limitations kindly suggested by the reviewers. While we recognize the importance of acknowledging study limitations, we believe that repeating them in the Conclusion may be redundant. Instead, we have ensured that the Discussion section provides a comprehensive and transparent account of the study’s limitations while keeping the Conclusion focused on summarizing key findings and policy implications. However, we are open to further revisions if the reviewers consider it necessary.	“A limitation of our research was the use of self-reported diagnosis; there is evidence of reasonable agreement between self-report and medical records. but the accuracy of self-report may be affected by factors such as severity and acuity of the disease and the possible overestimation of women or underestimation of men. Future studies should consider using multiple sources of information to determine the prevalence (rather than purely self-report methods) to reduce potential reporting bias. One key limitation of this study is the methodological differences between the three national health surveys used (ENSANUT Mexico 2018, ENSANUT Ecuador 2018, and PNS Brazil 2019). Each survey employed distinct sampling strategies, data collection methods, and weighting procedures, which may introduce variability in the comparability of the results. While we harmonized key variables (e.g., education, employment status, and chronic disease diagnoses) to ensure consistency across datasets, some differences may still persist. To enhance transparency, we have included a methodological comparison table

Comment	Response	Changes to the manuscript
		(Supplementary Materials) outlining key differences and similarities between the surveys. Another limitation of our study is the failure to include other determinants in the social environment such as discrimination or even the language spoken. Castro, Savage and Kaufman consider that an important factor for women to have worse health outcomes is due to discrimination on the part of health providers. This discriminatory factor has already been reported in other studies in LA. The real and perceived low economic level associated with dark-skinned people can also be a reason for discrimination by health service providers” (Please see page 15, lines 316 to 331, of the main manuscript).
Reviewer #3		
1. The paper needs a stronger motivation for the selection of these three countries. Even with the focus on Latin America there are tons of other data sources like DHS, MICS, STEPS and country health surveys (e.g., Argentina (ENNyS2), Chile (ENS), Colombia (ENDS) and Uruguay).	We appreciate the feedback and acknowledge the importance of health data sources from other Latin American countries, such as DHS, MICS, STEPS, and the national surveys mentioned. However, our decision to prioritize Brazil, Ecuador, and Mexico was based on several key factors that we consider essential for the proposed analysis. Firstly, these countries were selected based on the availability of harmonizable data on chronic diseases, gender, ethnicity, and socioeconomic conditions, which are critical for understanding health disparities in the region. While other countries like Argentina, Chile, Colombia, and Uruguay have valuable data sources, the three selected countries offer a combination of consistent and comparable data, enabling a more robust and effective analysis.	

Comment	Response	Changes to the manuscript
	Additionally, as mentioned, Brazil, Ecuador, and Mexico together represent over 50% of the Latin American population, making them highly relevant for studying regional health disparities. Including these countries provides significant coverage of the region's socioeconomic, ethnic, and geographic diversity, which is crucial for understanding health disparities across Latin America as a whole. We hope this explanation clarifies our choice and reinforces the relevance of the approach taken for the proposed analysis.	
2. The authors should justify pooling the observations of three countries that differ so much in the cultural underpinning of gender and ethnicity as well as health systems, social protection networks, economic and social development. (How much of the variation is explained by country fixed effects?)	Thank you for this thoughtful comment. We acknowledge that Brazil, Ecuador, and Mexico differ in cultural, economic, and healthcare system structures, which could influence the relationships between social determinants and chronic disease risk. The decision to pool observations across these three countries was guided by three key factors:  1. Shared Structural Health Inequities in Latin America: Despite national differences, systemic disparities in access to education, employment, and infrastructure disproportionately impact ethnic minorities and socioeconomically disadvantaged groups in all three countries. Pooling data allows us to identify generalizable patterns while also highlighting subgroup-specific trends. 2. Statistical and Methodological Considerations: We accounted for country-specific effects by including country fixed effects in our models, ensuring that our analysis controls for baseline differences in health systems, social policies, and economic contexts. 	

Comment	Response	Changes to the manuscript
	3. Regional Relevance and Representativity: By including Brazil, Ecuador, and Mexico, which together represent a significant portion of the Latin American population, our analysis provides insights that are highly relevant for the entire region. These countries are not only diverse in their own right but also offer a valuable cross-section of Latin American societies, providing a broader perspective on the social determinants of health that is applicable to other countries in the region. In response to this comment, we have updated the Methods section to explicitly justify our approach and report the contribution of the country as control variables in explaining variability in our models.	
3. Could the study be enriched by looking at information on STEPS or some other health surveys where there is self-reported and actual tests are performed (e.g., through blood tests or blood pressure tests).	Thank you for this valuable suggestion. We acknowledge that incorporating data sources such as STEPS or other health surveys that include both self-reported and clinically measured health indicators (e.g., blood tests, blood pressure tests) could further enrich the analysis and reduce potential biases related to self-reported diagnoses. However, our study focuses on nationally representative health surveys (ENSANUT Mexico 2018, ENSANUT Ecuador 2018, and PNS Brazil 2019) due to their large sample sizes and detailed social determinant variables that allow for a comprehensive analysis of gender, ethnicity, and infrastructure access in relation to chronic disease risk. While STEPS and other biomarker-based surveys provide valuable clinical data, they often have smaller sample sizes and may not always include detailed socioeconomic and demographic variables, limiting their ability to capture the broader structural determinants of health disparities. Additionally,	

Comment	Response	Changes to the manuscript
	harmonizing clinical and survey-based data across multiple countries presents significant methodological challenges due to differences in measurement techniques, target populations, and available indicators. Nonetheless, we recognize the importance of integrating objective health measurements into future research.	
4. While the paper offers a theoretical framing of gender as a socially constructed set of roles, relationships, and power dynamics, it fails to explain how this complex concept was operationalized in the empirical analysis. Despite using data from three different national health surveys in Brazil, Ecuador, and Mexico, the paper provides no information about how gender was coded in these surveys, whether coding was consistent across countries, if gender was self-reported or assigned, or how non-binary gender identities were handled. This omission creates a significant gap between the paper's theoretical framework and its empirical implementation. Understanding the exact measurement of gender is crucial for interpreting the findings about gender differences in health outcomes and social determinants, especially given potential cross-cultural variations in gender conceptualization across Latin America.	Thank you for this insightful comment. We recognize the importance of clearly explaining how gender was operationalized in our empirical analysis, ensuring alignment with our theoretical framework. In response, we have updated the Methods section to provide a detailed explanation of how gender was coded in the three national health surveys (ENSANUT Mexico 2018, ENSANUT Ecuador 2018, and PNS Brazil 2019). In all three surveys, gender was self-reported by respondents. However, the available data only included binary categories (man and women), without explicit options for non-binary or gender-diverse identities. We acknowledge this limitation, as it restricts our ability to capture the full spectrum of gender identities and may not fully reflect the complex, socially constructed nature of gender discussed in our theoretical framework. Despite this constraint, we ensured that gender coding was harmonized across the three surveys to maintain consistency in our analysis. We also recognize that cultural differences in gender conceptualization across Latin America may influence how individuals perceive and report their gender identity, which is an important consideration when interpreting our findings.	“In the three surveys we used, gender was obtained from the self-report of the sex of each respondent.” (Please see page 5, lines 106 to 107, of the main manuscript).

Comment	Response	Changes to the manuscript
	To enhance transparency, we have revised the manuscript to explicitly state how gender was measured, coded, and harmonized across countries.	
5. The paper's treatment of ethnic classification raises methodological concerns. While it analyzes ethnic disparities across four broad categories (mixed ethnicity, indigenous, black, and "others"), it fails to explain how these categories were constructed and harmonized across Brazil, Ecuador, and Mexico - three countries with distinct ethnic classification systems and historical contexts. The paper does not specify whether ethnicity was self-reported or assigned, nor does it justify critical methodological decisions such as combining mulatto, montuvio, and whites into an "other" category. This is particularly problematic given Latin America's complex racial and ethnic hierarchies and the varying meanings of ethnic categories across countries.. Without a clear explanation of how ethnic categories were operationalized and standardized, it becomes difficult to interpret the reported ethnic disparities in health outcomes and evaluate the broader implications of the study for understanding health inequities in Latin America.	Thank you very much for your observation. Ethnicity is an extremely important variable in our analysis, so clarifying its classification and construction is of utmost importance to understand the implications of our study. In Brazil and Ecuador, ethnicity is self-reported by respondents, and the ethnic category used in the analysis are directly derived from their self-assigned responses. This self-identification method reflects the cultural context and allows individuals to choose the ethnic category that best represents their identity. However, we acknowledge that there are differences in the ways ethnicity is conceptualized across countries, which is an important factor to consider when interpreting the results. For Mexico, the absence of a direct self-assignment question for ethnicity in the 2018 survey led us to rely on the existing ethnic classification system. While we recognize the importance of aligning classifications across countries, this was a necessary adjustment to harmonize the dataset while considering the limitations of the available data. Regarding the classification of “mulattoes”, “montuvios” and “white” individuals into the "Other" category, we note that these groups are often classified similarly in local studies (28) on health outcomes have created similar classifications mainly because they are groups with a small percentage compared in the total population and shared similar socioeconomic characteristics. This categorization reflects how ethnic	

Comment	Response	Changes to the manuscript
	groups with smaller population sizes and similar socio-economic conditions are typically grouped in both regional studies and national surveys. We believe that these decisions were necessary for the harmonization of ethnic categories, while still respecting the complexities of ethnic identification in each country. We have updated the manuscript to clarify how ethnicity was operationalized and harmonized in this study, to ensure transparency in the methodology and to allow for a more accurate interpretation of ethnic disparities in health outcome.	
6. Are there enough observations to make reliable inference in each of the ethnicity/gender cells in each country?	Thank you very much for your comment and for your concern about the inferences. In this sense, we used data from the total population, applying the expansion factors specific to each country. The observations in each ethnicity/gender group were weighted to reflect the proportion of each group within the total population of each country. Subsequently we obtained a random sample of 170,000 observations from each group, ensuring that there were no statistical differences between the samples and the total population of each group. In this sense, the same number of observations were used for training the random forest model for each group. In addition to this, we trained and tested the models with different datasets (training data and test data). This helps ensure that these models do not overfit and can only provide information and knowledge about the sample with which they were trained. Given these procedures, we consider that there is sufficient evidence to make reliable inferences for the entire population of each ethnicity/gender group in the three countries analyzed.	

Comment	Response	Changes to the manuscript
7. The paper doesn't do a sufficient job explaining why machine learning methods are preferable for this analysis. The only real justification appears on page 3, lines 53-55: "Machine learning can reveal linear and non-linear relationships and often outperforms traditional models." It would be useful to compare the results with more traditional approaches.	Thank you for your insightful comment. We appreciate the importance of comparing machine learning methods with traditional approaches to strengthen the justification for their use. While the primary objective of this study is not to compare modelling approaches, we have conducted a logistic regression analysis for comparison purposes. The results, which are presented in the Supplementary materials (Table S5), indicate that, in general, the machine learning model outperforms the traditional logistic regression in terms of predictive accuracy.	
8. The paper says that they have more than 96 million individuals. Is this because they use the expansion weight? If this is the case, it will be useful to describe the statistical procedures in terms of individual survey observations without using expansion weights.	Thank you very much for your suggestion and your question about the number of individuals. In this regard, we use expansion factors (weights) to obtain the estimate of the total adult population in the three countries before performing the descriptive analyses and machine learning predictive models. These expansion factors allow us to ensure that the sample accurately represents different age groups, sexes, genders, ethnicities, and socioeconomic conditions, avoiding any over- or under-representation. We agree that performing analyses without taking these weights into account could lead to biased estimates. For this reason, we use the weights solely for obtaining population-level estimates and ensure that they are not included in the machine learning models or the results presented. This approach allows us to maintain the integrity of the statistical analyses while providing valid, population-representative estimates.	
9. The use of expansion weights allows researchers to generate population-representative statistics. However, their application in	We appreciate your comment on the use of expansion weights. We recognize that these weights, designed for	

Comment	Response	Changes to the manuscript
machine learning contexts, particularly for measures of model fit and variable importance, raises methodological concerns. The core issue is that expansion weights, which replicate individual observations to match population totals, can distort the assumptions underlying machine learning algorithms. Random forests and similar methods assume independent observations and rely on actual data variation to assess feature importance and model fit. Using weighted observations could artificially inflate model performance metrics and bias variable importance measures by creating dependencies in the data. The authors need to address more explicitly how these survey design elements are incorporated into the analysis and clearly distinguish between using weights for population inference versus model estimation.	population inferences, could distort the assumptions of machine learning algorithms. Therefore, we use expansion weights primarily for descriptive analyses, ensuring that statistics are representative of the population. However, we avoid using them during the training of our machine learning models, particularly Random Forest, as they assume independent observations. Instead, we employ random sampling to create a dataset of 170,000 individuals per ethnic group and gender, allowing the models to be trained with independent observations. To determine the importance of variables, we used an agnostic approach, rather than relying on specific Random Forest metrics. Specifically, we examined the decrease in AUC (area under the curve) when each variable was neutralized., which offers a more robust measure of feature relevance while reducing the risk of bias. Model performance was assessed using AUC, which is particularly well-suited for handling imbalanced data, as show in Janitza, Strobl & Boulesteix (29). This approach allows us to address our research questions while mitigating the risks of bias that could arise from using expansion weights In the model estimation phase, ensuring the validity of our findings in both population inference and predictive modelling.	
Minor comments: 10. Line 63, the word "Brazil" is missing.	Thank you for your comment, we have added "of Brazil" on line 63	
11. Line 72, I would suggest using "employment status" rather than "occupation" to describe formal/informal employment, unemployed, or retired categories.	Thank you for your suggestion regarding the name change on line 72. To make things clearer, we have added "or employment status" in parentheses so that	“For the independent variables, we obtained data at the individual level on whether the person's home had water, sewage, garbage collection, what occupation (or employment status) they had (formally employee,

Comment	Response	Changes to the manuscript
	readers more familiar with this term can better understand this variable.	informal employee, unemployed, and retired), and education (basic level, intermediate level and higher level).” (Please see page 4, lines 96 to 99, of the main manuscript).

References

1. Braverman-Bronstein, A. *et al.* Gender inequality, women's empowerment, and adolescent birth rates in 363 Latin American cities. *Soc Sci Med* **317**, 115566 (2023).
2. Ribeiro, F. S., Crivelli, L. & Leist, A. K. Gender inequalities as contributors to dementia in Latin America and the Caribbean: what factors are missing from research? *Lancet Healthy Longev* **4**, e284–e291 (2023).
3. Abramo, L., Cecchini, S. & Ullmann, H. Enfrentar las desigualdades en salud en América Latina: el rol de la protección social. *Cien Saude Colet* **25**, 1587–1598 (2020).
4. Batis, C., Mazariegos, M., Martorell, R., Gil, A. & Rivera, J. A. Malnutrition in all its forms by wealth, education and ethnicity in Latin America: who are more affected? *Public Health Nutr* **23**, s1–s12 (2020).
5. Mesenburg, M. A. *et al.* Ethnic group inequalities in coverage with reproductive, maternal and child health interventions: cross-sectional analyses of national surveys in 16 Latin American and Caribbean countries. *Lancet Glob Health* **6**, e902–e913 (2018).
6. Granados, Y. *et al.* Inequity and vulnerability in Latin American Indigenous and non-Indigenous populations with rheumatic diseases: a syndemic approach. *BMJ Open* **13**, e069246 (2023).
7. *US health in international perspective: Shorter lives, poorer health.* (National Academies Press, 2013).
8. Gordon, E. H. & Hubbard, R. E. Do sex differences in chronic disease underpin the sex-frailty paradox? *Mech Ageing Dev* **179**, 44–50 (2019).
9. Thompson, O. & Ajayi, I. Prevalence of Antenatal Depression and Associated Risk Factors among Pregnant Women Attending Antenatal Clinics in Abeokuta North Local Government Area, Nigeria. *Depress Res Treat* **2016**, 1–15 (2016).
10. Crimmins, E. M., Kim, J. K. & Solé-Auró, A. Gender differences in health: results from SHARE, ELSA and HRS. *Eur J Public Health* **21**, 81–91 (2011).
11. Phillips, S. P. Defining and measuring gender: A social determinant of health whose time has come. *Int J Equity Health* **4**, 11 (2005).
12. Szanton, S. L., Seplaki, C. L., Thorpe, R. J., Allen, J. K. & Fried, L. P. Socioeconomic status is associated with frailty: the Women's Health and Aging Studies. *J Epidemiol Community Health (1978)* **64**, 63–67 (2010).
13. Assari, S., Nikahd, A., Malekahmadi, M. R., Lankarani, M. M. & Zamanian, H. Race by Gender Group Differences in the Protective Effects of Socioeconomic Factors Against Sustained Health Problems Across Five Domains. *J Racial Ethn Health Disparities* **4**, 884–894 (2017).
14. Lee, C., Park, S. & Boylan, J. M. Cardiovascular Health at the Intersection of Race and Gender: Identifying Life-Course Processes to Reduce Health Disparities. *The Journals of Gerontology: Series B* **76**, 1127–1139 (2021).
15. Engelman, M. & Jackson, H. Health Disparities at Older Ages: How Race, Gender, and Employment History Influence the Development of Functional Limitation. (2015).
16. Hankivsky, O. Women's health, men's health, and gender and health: Implications of intersectionality. *Soc Sci Med* **74**, 1712–1720 (2012).

17. Bodenheimer, T., Chen, E. & Bennett, H. D. Confronting The Growing Burden Of Chronic Disease: Can The U.S. Health Care Workforce Do The Job? *Health Aff* **28**, 64–74 (2009).
18. Umberson, D., Williams, K., Thomas, P. A., Liu, H. & Thomeer, M. B. Race, Gender, and Chains of Disadvantage. *J Health Soc Behav* **55**, 20–38 (2014).
19. Perreira, K. M. & Telles, E. E. The color of health: Skin color, ethnoracial classification, and discrimination in the health of Latin Americans. *Soc Sci Med* **116**, 241–250 (2014).
20. Wurtz, H. Indigenous Women of Latin America: Unintended Pregnancy, Unsafe Abortion, and Reproductive Health Outcomes. *Pimatisiwin* **10**, 271–282 (2012).
21. Kino, S. *et al.* A scoping review on the use of machine learning in research on social determinants of health: Trends and research prospects. *SSM Popul Health* **15**, 100836 (2021).
22. Louppe, G., Wehenkel, L., Sutter, A. & Geurts, P. in *Proceedings of the 27th International Conference on Neural Information Processing Systems* **1**, 431–439 (Curran Associates Inc., 2013).
23. Lundberg, S. M. & Lee, S.-I. in *Proceedings of the 31st International Conference on Neural Information Processing Systems* 4768–4777 (Curran Associates Inc., 2017).
24. Biecek, P. & Burzykowski, T. *Explanatory Model Analysis*. (Chapman and Hall/CRC, 2021). doi:10.1201/9780429027192
25. Hennis, A. J. M. *et al.* COVID-19 and inequities in the Americas: lessons learned and implications for essential health services. *Revista Panamericana de Salud Pública* **45**, 1 (2021).
26. Islam, N. *et al.* Social inequality and the syndemic of chronic disease and COVID-19: county-level analysis in the USA. *J Epidemiol Community Health (1978)* **75**, 496–500 (2021).
27. Rollston, R. & Galea, S. COVID-19 and the Social Determinants of Health. *American Journal of Health Promotion* **34**, 687–689 (2020).
28. Vásquez, M. A. *et al.* Prevalencia y nivel de concordancia entre tres definiciones de síndrome metabólico en la ciudad de Cuenca-Ecuador. *Avances en Biomedicina* **5**, 117–128 (2016).
29. Janitza, S., Strobl, C. & Boulesteix, A.-L. An AUC-based permutation variable importance measure for random forests. *BMC Bioinformatics* **14**, 119 (2013).

Manuscript Title: Decoding health disparities: gender, ethnicity, & chronic diseases in Latin Americans with individual data & machine learning

Manuscript ID: NCOMMS-24-63370B

While most of my previous comments were addressed (thanks for the answers), there are some remaining concerns, that are even reinforced by the authors. The first one related to the number of observations and use of survey weights. Authors indicate that “The total sample size collected across the three national surveys is approximately 170,000 individuals for each gender/ethnic group”, that is unlikely as the total number of observations is about that number, as the samples for the surveys are not expected to collect the same number of observations for the mentioned groups. It seems that authors are using weights as multipliers. A “sample” could include several identical observations if that is the case.	Thank you for informing us about this. We recognize that our approach was problematic. Our original approach was incorrect since we had used survey weights as simple multipliers to create a pseudo-sample of 170,000 observations per gender-ethnicity subgroup. We recognize that this approach may introduce some biases associated with the variability of the subsamples obtained and the possibility of repeated observations. Therefore, we have redone all the analysis. In the new analysis, we do not expand the sample. We use the original unweighted sample (Mexico = 42,068; Ecuador = 96,111; Brazil = 40,088; total N = 178,267) and supply each observation’s official expansion factor as a case weight directly in model training and the test. These changes implied to use a new implementation in our analysis that natively supports sample weights. This approach avoids duplicated observations and preserves the original sample structure. We have removed the statement about “170,000 per subgroup” and clarified the role of weights throughout.	Methods updated, line 96-97: “The actual number of respondents included in our analysis was 42,068 for Mexico, 96,111 for Ecuador, and 40,088 for Brazil.” Methods updated, line 131-133: “We use the expansion factors directly in the training of the algorithms and in cross-validation, and also directly in testing and model-agnostic approximation. We used the caret library²⁴ for cross-validation and the ranger library²⁵ to train the models with the best hyperparameters found, both in R 4.4.0 software.”
The second is related to the analysis; while I see the advantages of the approach, it is unclear how potential interactions are handled in	Thank you for pointing this out.	

the analysis. Individuals with a given level of education may also be more likely to have a specific type of employment or a differential probability of access to services. As far as I understand, random forest selects only some variables as the nodes, so may not include relevant interactions in each tree.	Indeed, some variables, such as education, can increase the probability of others in the people surveyed. In this sense, the random forest model implicitly captures interactions since it sequentially selects predictors to perform the classification. This creates decision paths where more than one variable interacts linearly or nonlinearly.	
The previous comments on the differences in NCD prevalence for Mexico and Ecuador's data remain unanswered. If I understand correctly, authors are using for Ecuador the Base 1 from the ENSANUT, with the question about a health issue in the previous 30 days. This does not seem comparable at all to the data from Mexico and Brazil.	Thanks for raising this, the reviewer is right to flag comparability across surveys. In our data construction we used different question wordings because the instruments differ by country, and we'll make that clearer in the manuscript. For Mexico and Brazil, the dependent variable is the respondent's self-reported diagnosis "by a health professional" of diabetes, cardiovascular disease, kidney disease, cerebrovascular disease, or obesity. For Ecuador, ENSANUT 2018 only captures whether respondents reported selected chronic conditions "in the last 30 days," so we used that item and treated a positive report as evidence of having the condition (on the assumption that knowing about a chronic condition implies prior contact with a health professional). This is described in the Analytical variables section, but we agree it should be more prominent. Conceptually, our comparative aim is not to estimate cross-country prevalence but to study how social determinants predict the presence of chronic disease at the time of survey. We therefore included country as a control variable in all models and focused inference on variable importance patterns by gender–ethnicity, rather than on direct prevalence comparisons across countries. This approach, together with our variable harmonisation, limits the risk that differences in question wording drive our main conclusions.	Discussion updated, line 333-353: “A key limitation is imperfect outcome harmonisation across surveys. Mexico and Brazil use “ever diagnosed” items, whereas Ecuador provides a 30-day report; these differences in recall window and wording mean cross-country prevalence levels are not directly comparable. We mitigate this by focusing interpretation on within-country patterns (gender /ethnicity strata) and by including country indicators in pooled models, but residual non-comparability is possible. Another limitation is that we do not adopt biomarker-defined outcomes in the main pooled analysis because comparable biomarker data are not available with full adult coverage in the same wave for all three countries, and relying on subsamples would substantially reduce effective sample sizes and introduce selection differences across settings. Future extensions will incorporate additional surveys and waves (including those with harmonised biomarkers) when they jointly provide nationally representative coverage of both objective health measures and the social-determinant variables required for our comparative framework. A further limitation is that the surveys capture sex (male/female) rather than gender identity or gendered social roles; accordingly, we use recorded sex as a proxy for gender. This proxy may introduce

	That said, we acknowledge the measurement is not perfectly identical across the three surveys. We already note methodological differences between ENSANUT Mexico 2018, PNS Brazil 2019 and ENSANUT Ecuador 2018 as a study limitation and have provided a methodological comparison in the Supplementary Materials; in the revision we will make this limitation more explicit with specific language about the Ecuador 30-day item and its implications for comparability, and we will clarify our harmonisation assumption in the Discussion.	misclassification and excludes transgender, non-binary, and other gender-diverse individuals, potentially biasing subgroup contrasts; the direction and magnitude of any such bias are uncertain and may vary by country and ethnicity. Relatedly, all outcomes are self-reported and thus susceptible to differential access to diagnosis, awareness, and reporting bias across subgroups (e.g., by education or service access). The net direction of any resulting bias is uncertain. Future work linking to clinical records or using harmonised biomarker-based definitions would improve comparability and strengthen causal interpretation, and incorporating harmonised measures of sex assigned at birth, current gender identity, and lived gender roles (with inclusive categories) would improve construct validity and inclusion.”
As the analysis is by sex and not by gender, it is essential to review the background and interpret the results accordingly. It is unclear how the report from Mexico and Brazil of ever diagnosed cases can be compared with the 30-day period in the Ecuador data.	Thank you for this helpful clarification. Our analysis and theoretical perspective are focused on gender, not sex. However, given that health surveys in the countries we analysed do not specifically measure or ask about gender, we used self-reported sex as a proxy for gender, assuming that self-reported sex is closely related to social roles, habits, and attitudes related to the identified gender. However, we are aware that this approach may exclude transgender, non-binary, or gender-diverse individuals. We have added a few lines to the study's limitations highlighting this fact. Regarding outcome comparability, Mexico (ENSANUT 2018) and Brazil (PNS 2019) record ever clinician-diagnosed chronic conditions, whereas	See modification above

	Ecuador (ENSANUT 2018) uses a last-30-days report of selected chronic conditions. We used the closest available indicator in each survey to capture the presence of chronic disease at the time of interview and harmonised these to a binary outcome for modelling. Because these wordings are not identical, we do not interpret cross-country differences in prevalence levels; instead, our pooled models include country indicators to absorb level shifts attributable to measurement, and our substantive interpretation focuses on within-country patterns by sex and ethnicity. We have also run country-specific models that reproduce the main patterns, which supports that our conclusions are not an artefact of the Ecuador 30-day window. In the revision we will make this limitation explicit in the Discussion that the Ecuador measure is not directly comparable to the ever-diagnosis items, so cross-country prevalence comparisons should be treated with caution.	
Still a more convincing argument is required on why those 3 countries. It seems to me that could be convenience, in the sense of previous knowledge of the datasets and about the context. Arguing that is due to harmonizable data assumes other surveys could not as harmonizable. While that could be true, there is no evidence provided on that. Having Brazil and Mexico plus almost any other country will result in a similar share of the regional population. It would also be important to recognize the real motive as a limitation.	Thank you for this helpful clarification. We tried to give you a better explanation about the selection of the countries. Our country set was defined a priori by the project’s scope, which targeted Mexico, Brazil and Ecuador (and initially Colombia) due to specific policy partners and planned knowledge translation in those settings. Within that remit, we applied transparent inclusion criteria—nationally representative adult surveys from 2018–2019 with microdata access; self-identified ethnicity/race including Indigenous and Afro-descendant groups; harmonizable chronic-disease and social-determinant measures; and published survey weights. Brazil, Mexico and Ecuador met these criteria within our timeframe. We recognise this is an analytic feasibility sample rather than a census of Latin America and note this as a limitation.	Methods (Data sources) sentence, line 79-82: “The study objective was to analyse Mexico, Brazil and Ecuador; surveys were included if they provided (i) nationally representative adult microdata from 2018–2019, (ii) self-identified ethnicity/race with Indigenous and Afro-descendant categories, (iii) harmonisable indicators of chronic disease and key social determinants, and (iv) public or timely licensable access.” Discussion (limitation) sentence, line 359-361: “Furthermore, our focus on three countries reflects the predefined scope and feasibility of the data; therefore, external generalizations beyond these settings should be made with caution until comparable data from other countries are incorporated.”

There is no answer to the specific question on the variation explained by country fixed effects.	Thank you very much for your concern about the country variable as a predictor in our analysis. We have included age and country as predictor variables in each of the trained models. However, given the focus on sex and ethnicity, we decided not to show the importance of these variables in predicting chronic disease diagnosis. However, we calculated significance as a percentage of the missing AUC for the country variable for each group. We have added the results of this analysis to the supplementary material. We also added a line in the results section to provide this information.	Results sentence, line 206-209: “Regarding the importance of the country variable, it was most significant among Indigenous women (7.88% of the missing AUC), followed by Indigenous men (6.65% of the missing AUC). On the other hand, it was least significant among men from the "other" ethnic group and Black men (2.56% and 1.4% of the missing AUC, respectively). Further details are provided in Table S2.”
Not clear on this answer. At least the Mexican ENSANUT included biomarkers, so that data is already available in the surveys you are pooling.	Thank you for this pointed this out. Indeed, ENSANUT Mexico includes biomarker modules. Our decision not to use biomarkers in the main pooled analysis was driven by comparability and coverage rather than convenience. The core aim was to estimate and compare how social determinants relate to chronic disease consistently across all three countries and across gender/ethnicity strata. To do that, we needed an outcome that is (i) available for the full adult sample in the same fieldwork wave, (ii) measured using closely aligned questions, and (iii) publicly accessible in all three datasets. While ENSANUT Mexico contains laboratory measures, comparable biomarker data are not uniformly available for the same wave and full adult sample in ENSANUT Ecuador 2018 and PNS Brazil 2019. Even within Mexico, lab modules typically cover subsamples, so adopting biomarker-defined outcomes would sharply reduce the effective sample especially within smaller sex/ethnicity strata and change the selection mechanism (participation/consent, fasting protocols, clinic subsampling), which risks introducing non-	Methods updated line 83-97: “We used a multi-country individual-level cross-sectional study with data from Mexico, Brazil and Ecuador. We extracted individual-level data from the National Health Survey of Mexico (“Encuesta Nacional de Salud y Nutrición” – ENSANUT, 2018), the National Health Survey of Ecuador (ENSANUT, 2018) and the National Health Survey of Brazil (“Pesquisa Nacional de Saúde” – PNS, 2019). The dependent variable is a binary indicator of chronic disease presence at interview, harmonised across surveys as follows: for Mexico and Brazil, we used self-reported clinician diagnosis (ever) of diabetes, cardiovascular disease, chronic kidney disease, cerebrovascular disease or obesity in all three countries. For Ecuador, the closest analogue available in the 2018 wave is the “last-30-days” report of selected chronic conditions; we coded a positive report as having the condition at interview. Although ENSANUT Mexico includes laboratory modules (e.g., glycaemia/HbA1c) and PNS measures blood

comparability and selection bias into the cross-country contrasts we report.

For these reasons, the primary outcome was harmonised to the element that all three surveys share with full coverage in the relevant wave: self-reported clinician diagnosis of chronic conditions (plus objectively measured BMI for obesity). This choice keeps the outcome definition consistent across countries and preserves statistical power for subgroup analyses. That said, we fully agree that objective measures are valuable. In the revision we will state explicitly in the Methods that we prioritised a single, fully comparable outcome across countries. We will also note in the Discussion that the absence of uniformly available biomarker data across countries is a limitation, and that future work will extend the pipeline to additional surveys (including STEPS) where comparable biomarkers and social determinants are jointly observed for nationally representative adult samples.

pressure, comparable biomarker data are not uniformly available for the same wave and full adult sample across all three countries, and laboratory modules are typically fielded on subsamples. To preserve cross-country comparability and statistical power in sex/ethnicity strata, our primary analyses therefore use the harmonised self-report definition. All pooled models include country indicators to absorb level differences attributable to measurement. The actual number of respondents included in our analysis was 42,068 for Mexico, 96,111 for Ecuador, and 40,088 for Brazil.”

Discussion updated, line 333-342:

“A key limitation is imperfect outcome harmonisation across surveys. Mexico and Brazil use “ever diagnosed” items, whereas Ecuador provides a 30-day report; these differences in recall window and wording mean cross-country prevalence levels are not directly comparable. We mitigate this by focusing interpretation on within-country patterns (sex/ethnicity strata) and by including country indicators in pooled models, but residual non-comparability is possible. Another limitation is that we do not adopt biomarker-defined outcomes in the main pooled analysis because comparable biomarker data are not available with full adult coverage in the same wave for all three countries, and relying on subsamples would substantially reduce effective sample sizes and introduce selection differences across settings. Future extensions will incorporate additional surveys and waves (including those with harmonised biomarkers) when they jointly provide nationally representative coverage of both objective health measures and the social-determinant variables required for our comparative framework.”

Surveys ask about sex not gender. This should be explicit and discussed how that limits the analysis related to gender.	Thank you for this important point. All three surveys record sex (male/female), not gender identity. We used self-reported sex as a proxy for gender given the more social focus of our study; however, we accept that this measurement may have limitations. This measurement limits the analysis in two ways: first, we cannot identify transgender, non-binary, or gender-diverse respondents, so our findings do not speak to those groups; second, observed differences by sex may reflect a mixture of biological factors and gendered social roles and exposures that our covariates only partially capture. We will make these constraints explicit and temper any language that could be read as inherently “gender-based” differences.	Discussion upgrade line 343-346: “A further limitation is that the surveys capture sex (male/female) rather than gender identity or gendered social roles; accordingly, we use recorded sex as a proxy for gender. This proxy may introduce misclassification and excludes transgender, non-binary, and other gender-diverse individuals, potentially biasing subgroup contrasts; the direction and magnitude of any such bias are uncertain and may vary by country and ethnicity”
Still not clear about the ethnic classification; how individuals from Mexico were coded? As not included in the survey in Mexico, the authors are assuming no blacks in Mexico? It would be useful to have a table with the descriptive data by country. Assuming census data, it could be well the case that you have mostly all blacks in Brazil, the majority of indigenous from Ecuador, and the majority of not sure if mixed or others from Mexico, depending on how non-indigenous from Mexico were coded.	Thank you very much for your interest and your question about ethnic classifications in Mexico. In this regard, indeed, and unfortunately, in Mexico, the 2018 ENSANUT does not directly measure ethnicity, so we use the variable that measures whether the person self-reports speaking an indigenous language. This way of measuring ethnicity in Mexico effectively leaves out the Black or Afro-descendant population living in Mexico. It wasn't until 2020 that Mexican censuses began to consider this ethnicity; however, it wasn't considered for the 2018 health survey. We are aware that this method of measurement may represent limitations and effectively leaves out the Black population in Mexico. We have added descriptive tables in the supplementary material by country. In the text, we clarify that cross-country contrasts involving Afro-descendant groups are limited to Brazil and Ecuador, and that Mexico	Method upgrade, line 121-123: “The questions and criteria used to measure diagnosed diseases and independent variables can be found in Table S1 in the supplementary materials.” Results upgrade, line 183-184: “Descriptive analyses by country are provided in Table S3-S5 of the supplementary material.” Discussion upgrade line 327-332:

	results should be interpreted as “speaks an Indigenous language vs other” rather than a full ethnicity taxonomy. We added a table in the supplementary material to describe the criteria we used per survey (see table S1).	“Another important limitation is that ethnic classification is imperfectly harmonised across surveys. In Mexico, ENSANUT 2018 does not capture self-identified ethnicity; we use “speaks an Indigenous language” as a proxy for Indigenous identification. This proxy may misclassify individuals and does not identify Afro-descendant people. As a result, cross-country contrasts involving Afro-descendant groups are limited to Brazil and Ecuador, and Mexican results should be interpreted as “Indigenous-language speakers vs other” rather than a comprehensive ethnicity taxonomy.”
Sampling weights were used to multiply observations. That seems not to be appropriate. Your statistical power is an artifact; you are treating a clustered sample as a simple random sample of an artificially homogenous population. Either you are using weights to create additional observations artificially or not using them at all. Your methods sections suggest that your random sample of 170k per group is from the 96 million individuals that result from creating new observations using the weights as multipliers. If this is the case, there is an issue here as observations are not independent at all.	Thank you very much for your concern about the use of weights, the expansion of the sample to the population, and their dependence. Our initial implementation used a widely adopted approach that did not support case weights in training. As a workaround, we created a pseudo-population by replicating observations according to expansion factors and then drew a self-weighted random sample for model training. As we mentioned before, we completely agree that this approach could lead to some biases in our results. We have updated all the analysis in this study. In the new analysis, we switched the approach and that implied to change the implementation in our analysis. Therefore, we adopt a new implementation that naturally supports case (survey) weights. We now fit models on the original survey samples and pass official expansion factors as weights directly in training, using cross-validation and a standard train/test split (without replication). We re-ran the entire analysis pipeline (pre-processing, tuning, model fitting, performance, and variable importance) and updated all tables, figures, and supplementary materials. Results are substantively unchanged: predictive performance and main effects are stable;	

	differences occur mainly among lower-importance predictors, which is consistent with removing pseudo-replication and random variation from re-sampling.	
--	---	--